# Hemocyanin facilitates lignocellulose digestion by wood-boring marine crustaceans

Katrin Besser[1], Graham P. Malyon [2], William S. Eborall[1], Giovanni Paro da Cunha[3], Jefferson G. Filgueiras[3], Adam Dowle[4], Lourdes Cruz Garcia[2], Samuel J. Page[5], Ray Dupree[5], Marcelo Kern[1], Leonardo D. Gomez [1], Yi Li[1], Luisa Elias[1], Federico Sabbadin[1], Shaza E. Mohamad[1,6], Giovanna Pesante [1], Clare Steele-King[1], Eduardo Ribeiro de Azevedo[3], Igor Polikarpov [3], Paul Dupree [7], Simon M. Cragg [2], Neil C. Bruce [1] & Simon J. McQueen-Mason[1]

Woody (lignocellulosic) plant biomass is an abundant renewable feedstock, rich in polysaccharides that are bound into an insoluble fiber composite with lignin. Marine crustacean woodborers of the genus *Limnoria* are among the few animals that can survive on a diet of this recalcitrant material without relying on gut resident microbiota. Analysis of fecal pellets revealed that *Limnoria* targets hexose-containing polysaccharides (mainly cellulose, and also glucomannans), corresponding with the abundance of cellulases in their digestive system, but xylans and lignin are largely unconsumed. We show that the limnoriid respiratory protein, hemocyanin, is abundant in the hindgut where wood is digested, that incubation of wood with hemocyanin markedly enhances its digestibility by cellulases, and that it modifies lignin. We propose that this activity of hemocyanins is instrumental to the ability of *Limnoria* to feed on wood in the absence of gut symbionts. These findings may hold potential for innovations in lignocellulose biorefining.

[1] Centre for Novel Agricultural Products, Department of Biology, University of York, York YO10 5DD, United Kingdom. [2] School of Biological Sciences, University of Portsmouth, Portsmouth PO1 2DY, United Kingdom. [3] Institute of Physics of São Carlos, University of São Paulo, 13566-590 São Carlos, Brazil. [4] Bioscience Technology Facility, Department of Biology, University of York, York YO10 5DD, United Kingdom. [5] Department of Physics, University of Warwick, Coventry CV4 7AL, United Kingdom. [6] Malaysia Japan International Institute of Technology, University of Technology, Malaysia, 54100 Kuala Lumpur, Malaysia. [7] Department of Biochemistry, University of Cambridge, Cambridge CB2 1QW, United Kingdom. Correspondence and requests for materials should be addressed to N.C.B. (email: neil.bruce@york.ac.uk) or to S.J.M.-M. (email: simon.mcqueenmason@york.ac.uk)

lignocellulosic, woody tissues provide mechanical support to higher plants. They are formed of a macromolecular composite of cellulose microfibrils coated and bound to one another by hemicellulose and lignin, producing a strong material that is resistant to biological and chemical degradation[1]. Cellulose is made of β-1,4-linked glucan polymers that form strong para-crystalline microfibrils, whereas the hemicellulose may be galactoglucomannans in softwoods (produced by gymnosperms such as pine) or predominantly glucuronoxylans in hardwoods (produced by angiosperms such as willow)[2]. These polysaccharides typically comprise ~70% of the wood, but are rendered hard to access within a hydrophobic macromolecular material interpenetrated and bound by the polyphenolic lignin. Lignin is unusual among biological polymers in that it is not made by specific polymerases. Instead, the constituent monolignols are secreted into the cell wall space and then polymerize without enzymatic assistance in situ by oxidative coupling, followed by a re-aromatization reaction, which can also form bonds between lignin and hemicellulose[3]. This gives rise to a polymer without a regular repeat structure to which specific lytic enzymes can evolve, resulting in high resistance to degradation.

Lignocellulose represents the largest pool of fixed carbon in the terrestrial environment and the abundance and availability of lignocellulosic materials such as timber, crop residues, and dedicated biomass crops makes them an attractive feedstock for the production of renewable fuels, chemicals, and materials without competing with food demand[4]. Despite their great abundance, the recalcitrance of woody materials to degradation presents a challenge for cost-effective biofuels production[1,5], as well as for digestion by heterotrophic organisms that feed on this material, such as fungi, bacteria, invertebrates, and other animals[6–9].

Fungi are preeminent among terrestrial wood-degrading organisms, owing to their ability to degrade lignin. Fungal lignin degradation follows two general strategies: white-rot fungi such as *Phanerochaete chrysosporium* are able to fully depolymerize and metabolize lignin, whereas brown-rot fungi such as *Postia placenta* modify lignin extensively through depolymerization and repolymerization, but do not metabolize it, in order to gain access to the polysaccharides in wood[7,10–13]. These different strategies are reflected in the secretome of such fungi: white rots invest in an extensive array of ligninolytic peroxidases and carbohydrate-active enzymes (CAZymes), whereas brown rots use a radical-generating system based on Fenton chemistry to attack lignin and polysaccharides, and deploy a less extensive range of CAZymes to degrade wood[7,10,11].

Besides the well-studied capability of bacteria to degrade cellulose and hemicellulose, there is growing evidence of their ligninolytic capacity by the utilization of peroxidases, laccases, and superoxide dismutases[14,15].

A number of invertebrates have evolved to live on woody plant materials, particularly insects such as termites and beetles[9,16,17]. These animals typically have complex digestive systems, involving several stages of digestion, and rely on populations of specialized microbes resident in the digestive tract[9,16,17]. In this context, termites have been best studied and their digestive systems are home to a fascinating diversity of bacteria, archaea, and protists that contribute to highly efficient lignocellulose digestion, whereas modified lignin accumulates in the feces of lower and higher termites without being metabolized by the animal[9,16]. However, there is limited information about the mechanism for the lignin deconstruction in termites and beetles[6,16–18].

Although lignocellulose is produced by land plants, large quantities enter the marine environment through estuaries, mangroves, and salt marshes, providing a niche for specialist marine arthropods and mollusks[6]. Teredinid mollusks (shipworms) consume large amounts of wood and rely, at least partly, on endosymbiotic gill bacteria for lignocellulose digestion[6,19]. In contrast, marine isopod crustacean woodborers of the Limnoriidae (colloquially known as gribble) do not rely on microbial symbionts, and have digestive systems devoid of resident microbiota[6,20,21]. Recent work has begun to uncover the genes and enzymes involved in cellulose degradation in *Limnoria spp.*[21,22]. However, the mechanisms for overcoming the lignin barrier to allow these enzymes to access their substrates remain unknown.

King et al.[21] (2010) reported that hemocyanins are abundant in the digestive transcriptome of *Limnoria quadripunctata* and speculated that they might be involved in wood digestion. Hemocyanins are copper-containing proteins in the hemolymph of arthropods, in which they have evolved from structurally related phenoloxidases (POs)[21,23–25]. They are recognized as the major respiratory proteins in the hemolymph of Pancrustacea (Hexapoda and Crustacea) and share a type-3 di-copper active center involved in oxygen binding with POs, tyrosinases, and catechol oxidases[23–25]. Hemocyanins are shown to exhibit PO activity upon activation, which is thought to involve loosening of the tertiary structure enabling access to the copper-containing active site[24,26,27]. PO activity plays a role in immune responses, melanization, and sclerotization by generating quinones[24,27–29], which are cytotoxic and highly reactive. Quinones can undergo redox cycling, which can generate semiquinone radicals that form polymers or adducts during melanization and sclerotization, or reactive oxygen species (ROS) that have antimicrobial properties[30]. Previous work in crayfish showed hemocyanins to be involved in tanning the chitinous cuticle lining the hindgut during molting, but they are confined to gastroliths (temporary calcium storage organs) to fulfill this function[31].

Here, we present an investigation into the process of wood digestion in *Limnoria*, revealing the extent of lignocellulose breakdown and a characterization of the digestive proteome. We show that the oxygen-carrier hemocyanin is found in the site of wood digestion, has ligninolytic activity, and markedly increases the digestibility of wood by cellulases. We therefore suggest that hemocyanins play a key role in isopod woodborer digestion, adding another aspect to the multi-functionality of these proteins in arthropods.

## Results

**Wood transformation during digestion.** We carried out mass loss studies and compositional analyses comparing wood fed to *Limnoria* with their excreted fecal pellets (Fig. 1a) to gain insights into the lignocellulose digestion process. This showed that ~ 22% of ingested material is consumed during gut passage, which takes about a day, both for willow (angiosperm, hardwood) and Scots pine sapwood (gymnosperm, softwood) (Supplementary Table 1).

In willow, most, if not all, of this mass loss is associated with digestion of the partially crystalline cellulose fraction, of which more than half was removed, whereas the typically more accessible hemicellulose fraction, as well as lignin, accumulate in the fecal pellets and show little change in relative amounts when normalized to mass loss (Fig. 1b). Sugar composition analysis of the sequentially hydrolyzed polysaccharide fractions hemicellulose (using trifluoracetic acid, TFA) and cellulose (using sulfuric acid ($H_2SO_4$) following TFA hydrolysis) revealed that most of the mass lost during digestion was accounted for by release of glucose, the major sugar of the cellulosic fraction (Fig. 1c). Most of the small amount of TFA-resistant hemicellulose in wood was reduced in fecal pellets (either because the hemicellulose is degraded during digestion or becomes more susceptible to hydrolysis by TFA after digestion), as shown by the

marked decrease in xylose and mannose contents of the cellulosic fraction (Fig. 1c). In contrast, very little sugar of the hemicellulose fraction appeared to be mobilized, and consisted mainly of xylose and smaller amounts of glucose (Fig. 1d). This indicates that the mechanism of hardwood digestion is mainly targeting the cellulosic fraction, where most of the glucose is found.

A similar situation is evident when the animals are feeding on Scots pine, with the cellulosic fraction being reduced in mass by 40% (Supplementary Fig. 1a, b). In contrast to willow, the hemicellulosic fraction is also reduced in mass by ~ 20% (Supplementary Fig. 1a). Softwood hemicellulose consists mainly of galactoglucomannan, but also of some arabinoglucuronoxylan, and during digestion mostly mannose and some glucose are released from this fraction, with little loss of xylose content (Supplementary Fig. 1c). These data may reflect the ability of many cellulases to partially digest glucomannans which contain β-1,4-glucose units and suggest that polymers containing hexose sugars (mainly cellulose but also some glucomannan) are the main targets for softwood digestion by *Limnoria*.

Solid-state $^{13}C$ nuclear magnetic resonance (ssNMR) studies of willow wood and fecal pellets confirm the major loss of cellulose when spectra are normalized to lignin (Supplementary Fig. 2, inset). The spectra also reveal that there is little or no change in the ultrastructure of the residual cellulose following digestion, evidenced by the unaltered ratio of the C4 signals of interior-to-surface or ordered-to-disordered cellulose microfibril regions (89 vs 84 ppm) in wood versus fecal pellets (Supplementary Fig. 2).

Although direct assessments of chemical changes of single molecules in a heterogeneous complex such as lignocellulosic plant biomass cannot be accomplished by Fourier transform infrared (FTIR) spectroscopy alone, analyses of wood and fecal pellets corroborate our findings of the digestion process in *Limnoria* (Supplementary Fig. 3; Supplementary Table 2). Attenuated total reflectance (ATR)-FTIR spectra are consistent with an increase in hemicellulose (1160 and 1045 cm$^{-1}$), which may be less acetylated (1737 cm$^{-1}$), an increase in lignin (1630–1670, 1160, and 1140 cm$^{-1}$), which may be more oxidized (1640 and 1550 cm$^{-1}$), and a decrease in cellulose (985 and 896 cm$^{-1}$) in fecal pellets in comparison with wood (Supplementary Fig. 3a, b; Supplementary Table 2).

Taken together, these data suggest that hexose-containing polysaccharides (cellulose and glucomannan) represent the major target of the *Limnoria* digestive system, with little reduction in the lignin and pentose-containing hemicellulose fractions. This is supported by the dominance of cellulase-like glycosyl hydrolases (GHs) of GH families 7 and 9 found in this animal's digestive transcriptome[21]. In order to enable access of the cellulases to their substrate, the tight complex of cellulose with hemicelluloses and phenolic lignin must be disrupted. Hemicellulose connects with cellulose by interaction with the crystalline surface of micro-fibrils[32] and this association seems to be disrupted as indicated by the decreased amount of hemicellulose in the cellulosic fraction of the biomass (Fig. 1c; Supplementary Fig. 1b). As lignin is still present in the fecal pellets, this polymer presumably has been modified to disrupt the association with the polysaccharide fraction and to facilitate access to the cellulose. To further investigate the mechanisms of wood digestion, we examined the physio-chemical properties and proteome of the digestive system.

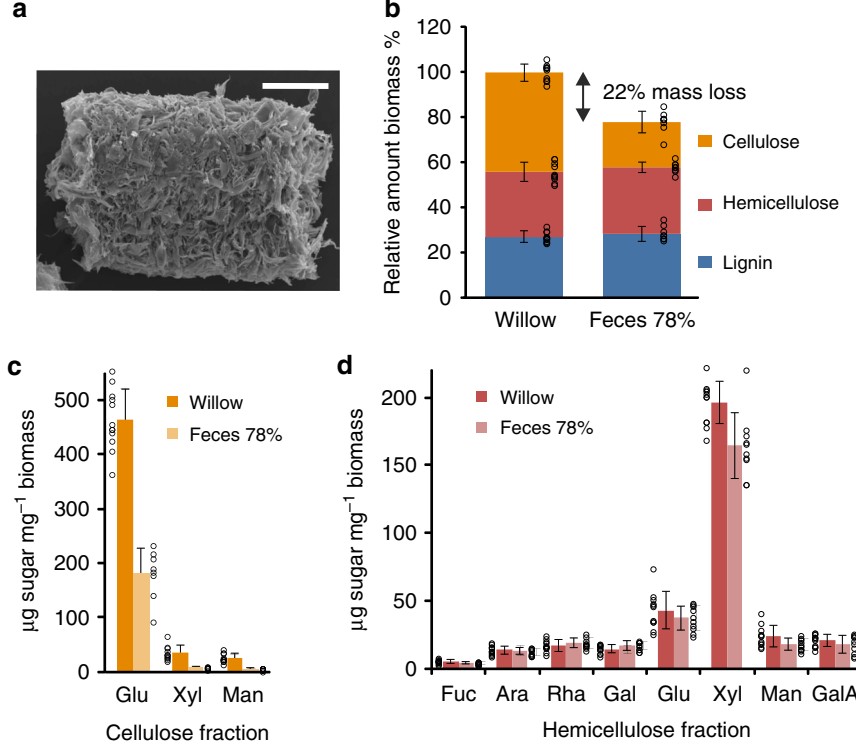

**Fig. 1** Major biopolymer composition of willow wood before and after digestion. **a** Scanning electron micrograph of fecal pellet from *Limnoria* (scale bar, 50 μm). **b** Relative amounts of biomass fractions from wood (Willow, N = 10) and fecal pellets (Feces 78%, normalized to mass loss during digestion, N = 8). Fractions comprise acetyl bromide-soluble lignin (blue), TFA-soluble hemicellulose (red), and sulfuric acid-soluble cellulose (orange). **c** Monosaccharide composition (absolute amount) of cellulose fraction from wood before (Willow, N = 11; dark orange) and after digestion (Feces 78%, normalized to mass loss during digestion, N = 8; light orange) analyzed by sequential $H_2SO_4$ hydrolysis following the TFA hydrolysis in **d**. **d** Monosaccharide composition (absolute amount) of hemicellulose fraction from wood before (Willow, N=11; dark red) and after digestion (Feces 78%, normalized to mass loss during digestion, N=10; light red) analyzed by TFA hydrolysis. Circles represent sample values and bars sample mean ± SD. Fuc, fucose; Ara, arabinose; Rha, rhamnose; Gal, galactose; Glu, glucose; Xyl, xylose; Man, mannose; GalA, galacturonic acid

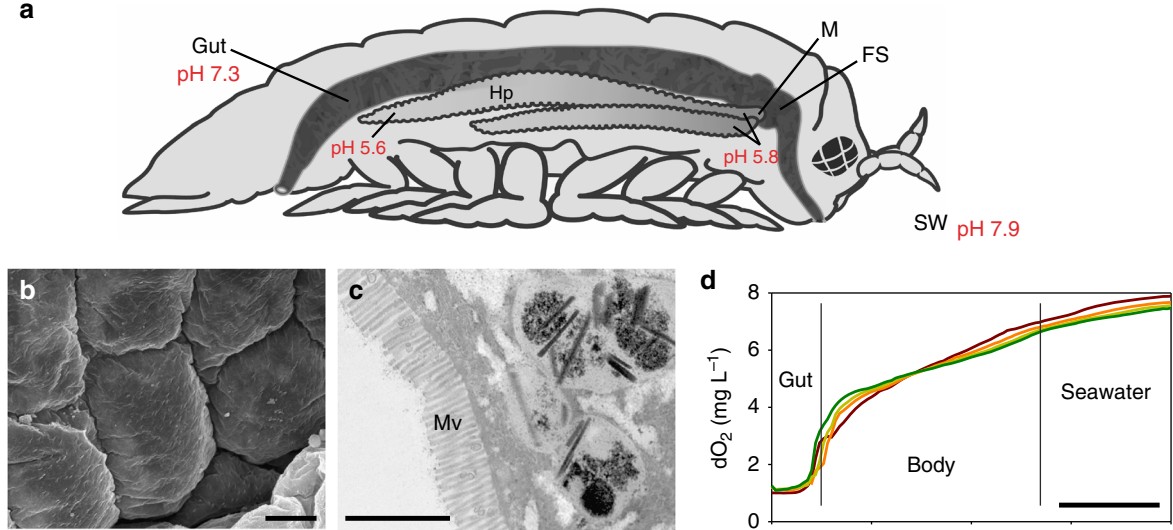

**Fig. 2** Digestive system of *Limnoria*. **a** Schematic lateral view of the animal (typically 2–3 mm long) showing one of the two bi-lobed hepatopancreases (Hp) connected to the anterior of the tubular gut; position of manifold (M) and filter system (FS) as indicated, food mass in hindgut indicated by dark coloration; pH values indicated at points measured. **b** Scanning electron micrograph of chitinous cuticle lining of the hindgut (scale bar, 5 μm). **c** Transmission electron micrograph of sectioned hepatopancreas cell with microvilli (Mv) facing the hepatopancreas lumen (scale bar, 2 μm). **d** Oxygen concentration measured by a microprobe withdrawn from the hindgut (Gut) through the body of four individual animals into surrounding seawater (scale bar, 0.5 mm)

**A two-compartment digestive system**. The digestive system of *Limnoria* is dominated by two structures: a cuticle-lined linear hindgut that is tightly packed with finely chopped wood particles and the hepatopancreas, which is the site of enzyme production and nutrient uptake (Fig. 2a–c)[21,22,33]. The hepatopancreas has a secretory and absorptive microvilli-lined epithelium facing its lumen that is kept free of wood particles by a complex filter system at the manifold connecting the hepatopancreas and the midgut, which leads into the hindgut (Fig. 2a–c)[33]. The hepatopancreas is contractile and, it is assumed, contracts to inject enzyme solutions, then relaxes, when the muscles of the hindgut contract to return breakdown products and enzymes for recycling, whereas spikes in the hindgut lining prevent the return of fecal matter. During feeding, seawater is ingested along with wood particles and is mixed with fluids in the digestive tract. This is evident in the pH gradient from the acidic hepatopancreas (distal region of lobes at pH 5.6 to proximal region of lobes at pH 5.8), to the neutral hindgut (pH 7.3), and to the exterior alkaline seawater (pH 7.9; Fig. 2a; Supplementary Fig. 4). The pH in terrestrial isopods is maintained ~ 6.0–6.5 in both organs, with a slight gradient from the most acidic distal end of the hindgut to the hepatopancreas[34], and similar trends and values were observed in freshwater amphipods[35]. We show by microprobe analysis that the hindgut has a low oxygen content compared with the rest of the body and the surrounding seawater, suggesting it is an oxygen sink (Fig. 2d). In the absence of gut-resident microbial populations this may reflect the action of an oxygen-consuming biochemical process rather than it being the result of microbial respiration, as suggested in other isopod digestive systems[34].

To identify the enzymes involved in the degradation of lignocellulosic biomass in the hindgut of the woodborer, we undertook proteomic analyses using label-free emPAI (exponentially modified protein abundance index)-based relative protein quantification of the fluids collected from the hindgut, with spectral data searched against a transcriptomic *Limnoria* library (PRJNA453115; SRP142516). A third of the proteins detected in the hindgut fluid comprise CAZymes (32.9%), which represent almost exclusively GHs and few carbohydrate esterases (0.8% of CAZymes) (PXD009486; MSV000082271). Known cellulase families GH7 and GH9 account for over 50% of the soluble GHs in the gut fluids (Fig. 3a). Hemocyanins, the oxygen-carrier proteins of crustaceans, were also detected to differentially accumulate in the gut fluids, as shown by comparing relative hemocyanin protein abundance in fluids and tissue/solid content of the hindgut (Fig. 3b). Reverse Transcriptase quantitative PCR (RT-qPCR) revealed that GHs and hemocyanins are specifically transcribed in the hepatopancreas, in which the proteins were detected by western analysis (Fig. 3c, d). The prediction of N-terminal signal peptides in GH and hemocyanin sequences (using SignalP 4.1, http://www.cbs.dtu.dk/services/SignalP/) suggested that proteins are secreted into the hepatopancreas lumen, from where they are moved into the hindgut, in which they accumulate as evidenced by immunoblots and proteomic data (Fig. 3c, d; Supplementary Fig. 5). The digestive proteome of *Limnoria* lacks peptides for known ligninolytic enzymes such as peroxidases, laccases, and other oxidoreductases known to degrade lignin[10,15] (PXD009486; MSV000082271).

**A role for hemocyanin in lignocellulose digestion**. Our study shows that respiratory hemocyanins are abundant and soluble in the hindgut luminal fluids of the marine woodborer, where they might play a part in wood digestion. Phenoloxidase activity of hemocyanins has previously been shown to occur in the hemolymph and in gastroliths of arthropods to fulfill multiple functions in immune response reactions and cuticle tanning[24,27–29,31]. To examine a potential role in wood digestion, native hemocyanins from *Limnoria* were purified by gel filtration and their identity confirmed by western blot analysis (Supplementary Fig. 6a, b). Analysis of the purified and concentrated fractions revealed a strong enrichment in hemocyanin, which forms multimers, most likely hexamers and multiples of these (Fig. 3e), as has been observed for terrestrial isopods and arthropods[23,28,36]. We tested the purified hemocyanin for potential oxidative activity on small phenolic compounds and found that it oxidized

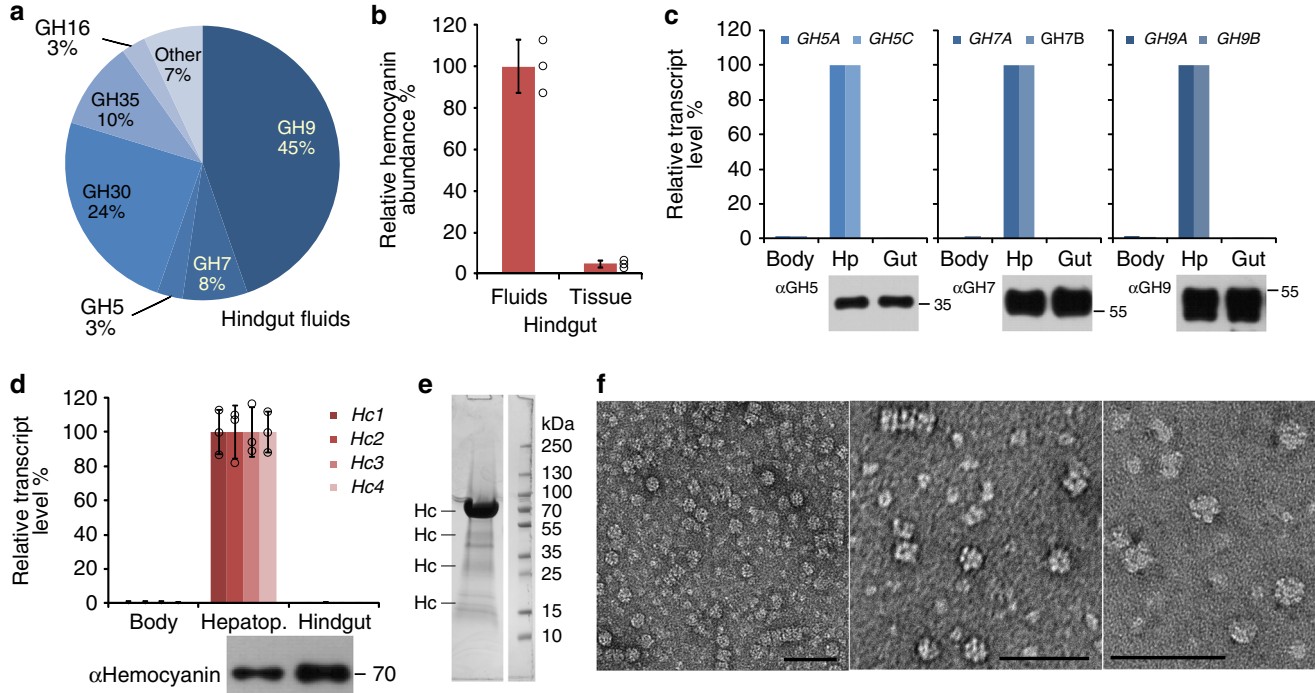

**Fig. 3** Transcripts and proteins in the *Limnoria* digestome and native hemocyanin extract. **a** Relative abundance of glycosyl hydrolase (GH) families detected in hindgut fluids based on molar percentage. Values represent sample mean, $N = 3$ ($N$ consisting of material from 100 animals each). **b** Relative hemocyanin abundance in hindgut separated into tissue (including solid content) and fluids by comparison of relative molar percentage of hemocyanins. Circles represent sample values and bars sample mean ± SD, $N = 3$ ($N$ consisting of material from 100 animals each), replicates were normalized intra-sample on the expectation of equal total protein amount between replicates when summed across all organs. **c** Top: differential relative gene expression of two members each of GH family 5, 7, and 9 cloned from *Limnoria* [GenBank accession numbers GU066826 (*GH5A*), GU066827 (*GH5C*), FJ940756 (*GH7A*), FJ940757 (*GH7B*), FJ940759 (*GH9A*), FJ940760 (*GH9B*)], measured by RT-qPCR: Body, remaining body part after removal of hindgut (Gut) and hepatopancreas (Hp). Values represent samples derived from pools of fifty organs each. Bottom: Spatial distribution of GH epitopes from families 5, 7, and 9 in hepatopancreas and hindgut, analyzed by western blots probed with anti-GH antibodies (αGH5, αGH7, αGH9). **d** Top: differential relative gene expression of four hemocyanin genes cloned from *Limnoria* [Hc1–4, GenBank accessions numbers GU166295 (*Hc1*), GU166296 (*Hc2*), GU166297 (*Hc3*), GU166298 (*Hc4*)] measured by RT-qPCR: Body, remaining body part after removal of hindgut and hepatopancreas (Hepatop.). Bar values represent sample mean ± SD, $N = 3$ ($N$ consisting of five organs each; individual sample values are only shown as circles for hepatopancreas). Bottom: spatial distribution of hemocyanin epitopes in hepatopancreas and hindgut, analyzed by western probed with anti-hemocyanin antibodies (αHemocyanin). **e** Coomassie-stained SDS–PAGE gel of native *Limnoria* hemocyanin in pooled and concentrated fractions from gel filtration (50 μg of protein) compared with molecular size marker in kDa. Hc, band containing hemocyanin as identified by protein ID. **f** Uranyl acetate negative stained TEM images of native *Limnoria* hemocyanin extract showing hexamers and stacked multiples of these (scale bar, 50 nm)

pyrogallol to form purpurogallin, as indicated by the increased absorbance at $\sim 320$ nm[37], and that this reaction required activation of hemocyanin with SDS (sodium dodecyl sulfate), but was independent of hydrogen peroxide supply (Fig. 4a, Supplementary Fig. 6c, d).

When hemocyanin was purified in seawater, the expected solvent in the woodborer's digestive system, its melting temperature ($T_m$) was lowered by 5 °C compared with the $T_m$ in buffer, indicating a slight unfolding of the protein (Supplementary Fig. 7a, b, c) sufficient to enable ligninolytic activity. Incubation of hemocyanin in seawater with alkali lignin led to distinctive modifications of the lignin structure, evidenced by ssNMR spectra (Fig. 4b, top panel). The most prominent change upon hemocyanin treatment is the intensity reduction in the spectral region corresponding to aromatic carbons (140–160 ppm), as well as in the signal attributed to aryl methoxyl carbons (56.1 ppm). With the exception of the C1 carbon of guaiacyl units, the spectral region between 140 and 160 ppm is caused by O-aromatic carbons (see caption of Fig. 4b). The intensity of the signal in the region of 110–140 ppm is also decreased and represents contributions from O-aromatic carbons. In addition, the spectrum of the hemocyanin-treated samples shows

a signal at 24.9 ppm, which is typical of $CH_3$ groups in aliphatic moieties. Although the NMR results alone cannot be specific about the exact mechanism of the hemocyanin-lignin interaction, the simultaneous relative reduction of the O-aromatic and aryl methoxyl carbon signals, as evidenced by the difference spectrum of hemocyanin- and seawater-treated samples (Fig. 4b insets), suggests that hemocyanin acts mostly on the aromatic-$OCH_3$ sites. This is also supported by the little or no change in the signal of ring carbons not linked to oxygen (105 ppm). Co-incubation of hemocyanin with the chelator diethylene triamine penta-acetic acid (DTPA) prevented these lignin modifications and NMR signals are very similar in both control and hemocyanin-treated samples (Fig. 4b, bottom panel), likely owing to hemocyanin inactivation as a result of copper sequestration from its active site by DTPA. In support of this, we found that addition of chelators to hemocyanin in seawater removed oxygen bound to its active site, evidenced by the loss of absorbance at $\sim 340$ nm (Supplementary Fig. 7d)[24,28]. It appears that chelator-dependent sequestration of copper from the hemocyanin active center, not only caused the loss of the oxygen binding property, but also led to further destabilization of the protein, as indicated by a significant temperature shift in the

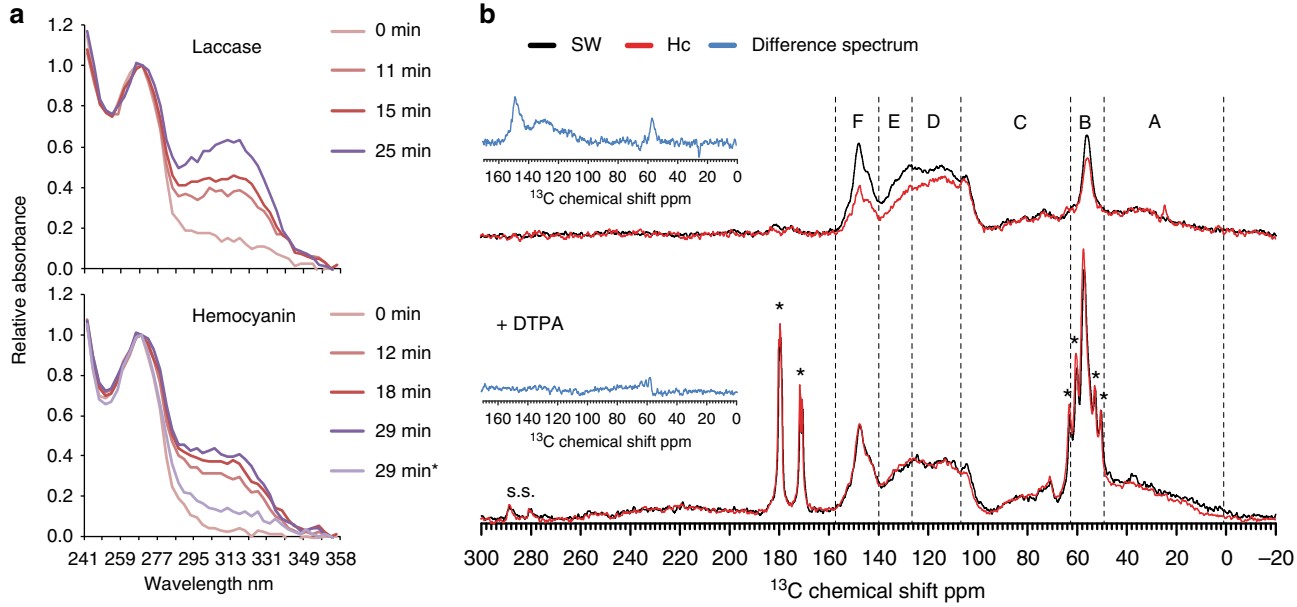

**Fig. 4** Hemocyanin activity on pyrogallol and isolated lignin. **a** UV-Vis spectra of *Trametes versicolor* laccase compared with activated (by addition of 2.75 mM SDS) hemocyanin in 0.1 M NaPO$_4$ pH 6.8 with 1 mM pyrogallol during incubation at room temperature for the time indicated (min; darker tones with increasing time). Asterisk * indicates spectrum for hemocyanin incubation without SDS addition (no activation, light purple). Absorbance maximum of pyrogallol at 267 nm and of oxidation product purpurogallin at ∼ 320 nm. **b** $^{13}$C solid-state NMR spectra of alkali lignin (3%) after 1 h incubation in seawater (SW, black) or with hemocyanin (Hc, red), top panel: activated Hc, bottom panel: Hc inhibited with chelator (50 mM diethylene triamine penta-acetic acid, DTPA). The spectra were normalized to match the signal of aliphatic (10–60 ppm) and O-aliphatic carbons (60–110 ppm) of lignin. The spectral regions delimited by capital letters correspond to: A—aliphatic carbons of lignin. The sharp peak at 24.9 ppm is characteristic for CH$_3$/CH$_2$ from saturated side chains; B—aryl methoxyl carbons of lignin; C—OC$_\alpha$H$_2$, OC$_\beta$H$_2$, OC$_\gamma$H$_2$ carbons of lignin; D—C2 and C6 aromatic carbons of syringyl, C2, C3 and C6 aromatic carbons of guaiacyl; E—C1 and C4 aromatic carbons of syringyl (e,ne), C1 aromatic carbons of guaiacyl; F—C3 and C5 aromatic carbons of syringyl and C4 aromatic carbons of guaiacyl. Chemical shifts were assigned based on literature data (Supplementary Refs 18–20). s.s. spinning sidebands; signals with asterisk * at (179–180 ppm) and (170–173 ppm) are from carbonyl carbons and at 51, 53, 60, and 63 ppm are from CH$_2$, attributed to free DTPA Z molecules[78], i.e., not forming Cu-DTPA complexes. Note that owing to its paramagnetic nature, Cu$^{+2}$ induces a fast relaxation of the $^{13}$C in its vicinity, so the signal of Cu-DTPA complexes is not expected to show in $^{13}$C CP-MAS spectra. Insets: difference spectra (blue) of hemocyanin- and seawater-treated lignin samples

melting curve with a $T_m$ of 70 °C to below 55 °C (Supplementary Fig. 7a, c), both potentially contributing to inactivation of the protein. It is interesting to note that chelators had no effect on the conformation and stability of (inactive) hemocyanins in buffer without SDS, as evidenced by an unshifted melting curve and stable $T_m$ with increasing chelator concentration, confirming that the copper in the active center is not exposed and therefore not accessible for sequestration or enzymatic activity in buffer without SDS (Supplementary Fig. 7b, c).

**Hemocyanins as effective pretreatment for saccharification.** The pretreatment of woody biomass is required to reduce its recalcitrance and to improve accessibility and digestibility of the polysaccharide fractions by hydrolytic enzymes (saccharification)[1,5]. In industrial processes this generally involves thermo- or physico-chemical processes, using acidic, alkaline, or organic solvents combined with high temperature and/or pressure[5], whereas wood-degrading organisms use a range of ligninolytic enzymes and/or Fenton chemistry to achieve this[6]. Our data suggest that hemocyanin in the *Limnoria* digestive system provides the phenoloxidative power for lignin disruption, improving access of cellulases to their substrate. To test this hypothesis we pretreated wood with hemocyanin prior to digestion with cellulases, and assessed the impact on its saccharification.

We found that incubation of powdered wood with *Limnoria* hemocyanin in seawater for short periods (10–20 minutes) at room temperature is sufficient to increase cellulase activity by 50–300% compared with controls without hemocyanin pretreatment (Fig. 5, Supplementary Fig. 8a). Similar results were obtained using the recombinant limnoriid *Lq*Cel7B (*Lq*GH7B), a processive cellobiohydrolase (CBH)[22], or the fungal *Hj*Cel7A (*Hypocrea jecorina* CBH I) in saccharification reactions (Supplementary Fig. 8b), as well as using softwood (Scots pine sapwood, Supplementary Fig. 8c) instead of the hardwood (willow), which have different lignocellulose compositions. This pretreatment effect of hemocyanin was suppressed by addition of the chelator DTPA, which inhibited hemocyanin activity on soluble lignin as shown above (Fig. 5a, Fig. 4b). Similarly, hemocyanin pretreatment in the presence of the strong reducing agent dithionite prevented increased saccharification by CBH, likely owing to the reduction of the enzyme's copper center (Fig. 5b)[38]. Hemocyanin incubation had only a minor impact on the digestibility of pure phosphoric acid swollen cellulose (PASC) and this was unaffected by chelator treatment (Supplementary Fig. 8d), indicating that the increased digestibility of wood does not arise from direct effects on cellulose. The small increase in cellobiose release from PASC by hemocyanin pretreatment is likely due to the activity of the small amount of cellulases co-purified with hemocyanin (GHs, Supplementary Fig. 6c).

One way that protein addition might enhance cellulase action on wood might be by reducing non-productive binding of cellulases to non-cellulosic components such as lignin[39,40]. To assess if such an effect might be responsible for the enhanced cellulase action, the binding of cellulases to hemocyanin-

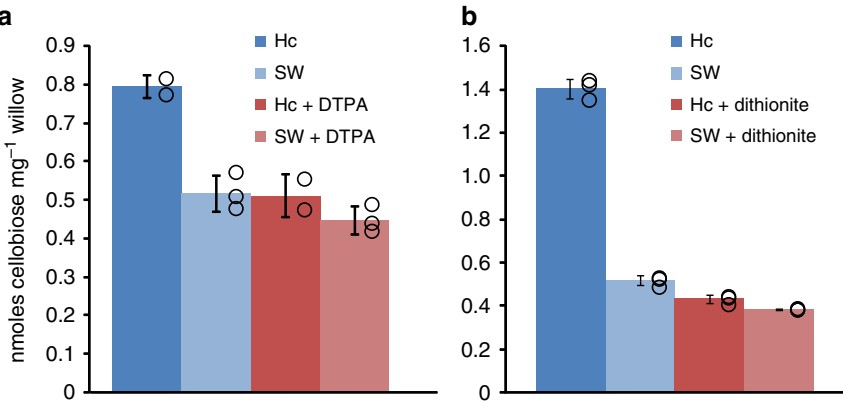

**Fig. 5** Digestibility of willow following short hemocyanin pretreatments at room temperature. Digestibility of willow measured as release of cellobiose (nmoles) per unit biomass (mg) in saccharification reactions upon biomass pretreatment. Pretreatment of willow with hemocyanin in seawater (Hc, dark tones) or seawater alone (SW, light tones) without (blue colors) and with ( +, red colors) 50 mM DTPA (chelator) for 10 min (**a**) or 50 mM dithionite (reducing agent) for 20 min (**b**), followed by saccharification with CBH I from *Hypocrea jecorina* (*Hj*CBH I). Hydrolysates were analyzed by HPAEC against cello-oligomer standards, and values normalized to reactions without CBH following pretreatments to account for any remaining cellulase activity derived from the hemocyanin preparation. Circles represent sample values and bars sample mean ± SD, *N* = 3 (*N* = 2 for Hc ± DTPA)

pretreated biomass was investigated. A similar proportion of the fungal *Hj*CBH I (containing a cellulose binding module) was found to bind to the biomass during saccharification reactions independent of hemocyanin pretreatment, as shown by Coomassie-stained SDS-polyacrylamide gel electrophoresis (PAGE) gels (Supplementary Fig. 9). A very small proportion of hemocyanin remained on the wood mass after removal of the pretreatment solutions, which is likely to be caused by protein being trapped in the biomass upon pelleting prior to removal (Supplementary Fig. 9). Therefore, it is unlikely that the enhancement of digestibility seen with active hemocyanin is caused by reduced non-productive binding of cellulases to the biomass.

To assess whether hemocyanin treatment led to increased porosity of wood enhancing access to cellulose by cellulases, we examined the $T_2$ relaxation time profiles of *N,N*-Dimethylacetamide (DMAc) in hemocyanin-treated wood using low field time domain NMR[41]. DMAc penetrates the pores in the material and its relaxation time $T_2$ becomes relative to the size of the pore it is filling (the smaller the pore, the shorter $T_2$) and to the pore surface ability to produce spin relaxation of the DMAc protons (so-called surface relaxivity parameter). This parameter depends on surface properties such as magnetic susceptibility and affinity to the enclosed fluid. Lignocellulose has characteristic interstitial scales (pores), presenting a $T_2$ profile with components that encode the various length scales of the pores as well as their occupancy by the enclosed fluid[42]. The proportion of DMAc in each interstitial scale is estimated from the relative area of the respective component of the $T_2$ distribution, which can be correlated to the accessibility of each type of pore by DMAc molecules.

When comparing willow wood treated with seawater or with hemocyanin, it is apparent that the $T_2$ profiles of DMAc in both samples are similar in shape, i.e., reveal the same type of pore distribution, with three pore populations at similar length scales (Supplementary Fig. 10a, b). Assuming a similar surface relaxivity of DMAc in willow as in cotton cellulose, the length scales corresponding to these profiles are estimated to be in the order of 1–10 nm, 10–100 nm, and 100–1000 nm (Supplementary Fig. 10b)[41]. These pore populations are interpreted as interstitial spaces in the amorphous region of cellulose (surface of cellulose fibrils), voids in the inter-microfibril spaces, and small luminal

structures, respectively[42]. However, the relative area of the two lower $T_2$ population curves increased considerably in the hemocyanin-pretreated sample (Supplementary Fig. 10b, c), which indicates that there are more pores within these two length scales (both associated with the accessibility of the cell wall) after hemocyanin incubation. This suggests that there is increased porosity of lignocellulose and potentially better accessibility of cellulose by hydrolytic enzymes upon hemocyanin pretreatment.

Liquid chromatography-tandem mass spectrometry (LC-MS/MS) analysis of the hemocyanin extract after trypsinolysis revealed that it contained low level contamination with ferritin (1.3%, based on emPAI-derived molar percentages; Supplementary Fig. 6c), which was also detected in the gut fluid proteome (0.5 mol%; PXD009486; MSV000082271). Therefore, we tested the effect of ferritin in pretreatment reactions on willow followed by saccharification with CBH to exclude the possibility that the ferritin co-extracted with hemocyanin in our experiments contributed to the observed digestibility improvement[43]. Ferritin pretreatment had no impact on digestibility of willow when used at a molarity five times higher than the one expected in the hemocyanin extract used in pretreatment reactions (Supplementary Fig. 11).

To compare the pretreatment performance of hemocyanin to other copper-containing phenoloxidases, *Trametes versicolor* laccase and *Agaricus bisporus* mushroom tyrosinase were used at the same molarity as hemocyanin (14 μM) in pretreatment reactions followed by saccharification and revealed no effect on digestibility by the laccase, and a smaller increase (than that by hemocyanin) for the tyrosinase (Supplementary Fig. 11).

We also show that the hemocyanin dependent enhancement of cellulase efficiency after ten minutes incubation at ambient temperature is comparable to mild thermochemical pretreatment with sodium hydroxide at 90 °C for 45 min (Supplementary Fig. 8a). To provide evidence for the potential recyclability of hemocyanins in view of industrial applications of ligninolytic agents, the pretreatment solutions from one experiment were removed after the incubation period and re-applied to untreated wood powder of a second cellulase digestion experiment. This repeated use of hemocyanin in pretreatment reactions still led to increased cellobiose release by cellulase from pretreated willow wood (Supplementary Fig. 8a).

## Discussion

We have studied the digestion of wood by crustacean woodborers of the genus *Limnoria* that have no detectable microbial symbionts in their guts[20–22]. Our data show that the digestive process mainly targets the cellulosic fraction, and also other hexose-containing polysaccharides of wood, through the action of endogenously produced cellulases of families GH7 and GH9 that are expressed in the hepatopancreas and constitute over 50% of soluble GHs in the hindgut fluids surrounding the wood. We found that hemocyanins are abundant in the digestive proteome of the woodborer, that they can modify lignin, and that pre-incubation of wood with hemocyanins leads to a marked increase in the digestibility of wood by cellulases, which is likely due to an increase in the porosity of the lignocellulose. Ander et al.[44] reported the activity of hemoglobin and other heme compounds towards lignin in the presence of oxidants such as hydrogen peroxide, in a similar fashion to peroxidase-type ligninases. In contrast, we report the discovery that respiratory hemocyanins in the *Limnoria* hindgut enhance lignocellulose digestibility by a mechanism that requires no additional oxidant and appears to function in a manner distinct to that reported for hemoglobin.

The considerable mass loss of 22% during wood digestion in *Limnoria* represents a loss of over 50% of the cellulose per day in willow (compared with few weeks for similar total mass loss of 10–50% by fungi[45]), leaving the fecal pellets enriched in the hemicellulose and lignin fractions. This differs from the degradation by wood-decay fungi, which first attack the more accessible polysaccharides (non-crystalline cellulose and hemicelluloses) before moving on to the crystalline cellulose fraction[10–12]. Xylophagous insects, such as beetles and termites, also usually degrade the hemicellulose fraction in addition to cellulose, aided by their microbial gut community[16,17]. In contrast to their herbivorous hosts, many symbiotic microbes are generally able to metabolize the pentose sugar xylose, resulting in nutritional benefits to the host upon release of these fermentation products[16,17]. In the absence of gut microbiota in the crustacean woodborer there may be no route to utilize xylose, explaining the accumulation of xylan in fecal pellets of *Limnoria*.

Cellulose occurs as partially crystalline microfibrils of β-1,4 glucans that GHs do not digest efficiently without facilitation through oxidative attack by free radicals or by enzymes like lytic polysaccharide monooxygenases (LPMOs)[6,7,11,46]. We could find no putative enzymes of classes recognized for oxidative polysaccharide degradation[7,11,46,47] in the digestive transcriptome or proteome of *Limnoria*. Our ssNMR data suggest that cellulose crystallinity is unchanged in fecal pellets, indicating indiscriminate degradation of amorphous and crystalline cellulose parts during wood digestion. The marine environment of the woodborer may aid the accessibility of cellulose in the absence of LPMOs, as the salts of seawater are thought to partially disrupt the hydrogen bonds between cellulose chains in microfibrils similarly to ionic liquids[48,49].

The lignin matrix in wood represents a major barrier to cellulose hydrolysis, hindering the access of cellulases to their substrates[1,5]. Wood-degrading fungi typically utilize extracellular reactive oxygen species for lignin breakdown and/or the action of oxidative enzymes (peroxidases and laccases)[6,7,10,11]. Lignin degradation to varying degrees has been reported in the microbe-containing gut systems of wood-feeding beetles, moths and termites, with modifications detected that resemble those caused by wood-decay fungi, including aromatic ring deconstruction and ring demethoxylation[18,50,51]. A combined host and symbiont transcriptomic approach in termites revealed that sequences encoding ligninolytic enzymes such as laccases were exclusively host-derived with only one putative peroxidase identified from symbionts[52]. Correlated laccase gene expression and phenoloxidase activity in the foregut and salivary glands, together with confirmation of phenoloxidase activity of laccases from the salivary glands towards lignin-phenolics, suggests a role for laccases in lignocellulose digestion by termites[52,53]. Transcriptomic profiling of animal and microbial digestive enzymes in wood-feeding beetles suggested a more cooperative approach to lignin degradation, and identified transcripts from both origins for reductases, dehydrogenases, laccases, other multi-copper oxidases, peroxidases, and other auxiliary lignocellulolytic enzymes[17,54].

Symbiotic as well as endogenous phenoloxidase activity has been observed in the digestive tracts of some peracarid (a group of crustaceans including isopods and amphipods) detritivores that feed, at least partly, on lignocellulose[55,56]. More recently, it has been proposed that *sensu strictu* phenoloxidases are lacking in peracarids and that in the hemolymph their function may be replaced by multifunctional hemocyanin, which is involved in respiration and immune responses[36]. Our work shows that hemocyanin from *Limnoria* is abundant inside the digestive system and able to oxidise pyrogallol, a lignin-derived phenolic, but also promotes aromatic ring cleavage of lignin. Lignin peroxidases and laccases involved in lignocellulose digestion in white-rot fungi have been shown to cleave side chains and aromatic rings of lignin model compounds[57,58], leading to efficient decomposition of lignin[7,10,58]. We speculate that in the absence of such proteins from gut fluids of the marine woodborer their ligninolytic function is provided by hemocyanin in the digestive tract, resulting in lignin modifications that improve accessibility of cellulases and leading to oxygen depletion of the hindgut.

We show that native *Limnoria* hemocyanins provide a highly effective pretreatment of wood, enabling cellulases to hydrolyze the cellulose fraction, whereas hemocyanin has no impact on the digestibility of cellulose itself. We also found that hemocyanin increased the porosity of the biomass. We, therefore, infer that it is the observed lignin-modifying activity that potentiates the hydrolysis of cellulose in wood. It is interesting to note that hemocyanin sequences have been identified in the digestive proteome and transcriptome of a termite and a xylophagous beetle, respectively, but no direct involvement in digestion has been shown[54,59]. When fungal phenoloxidases, implicated in lignocellulose degradation[58,60,61], were used under the same conditions as for hemocyanin in pretreatment reactions, it became apparent that the laccase had no impact on wood digestibility, maybe owing to the lack of a suitable mediator[60], whereas the mushroom tyrosinase increased digestibility. In this study, we have shown that mushroom tyrosinase potentially enhances enzymatic saccharification of wood and it is interesting to note that it shares the same type-3 copper center with hemocyanins. In addition to their suggested role as redox partner for LPMOs[62], tyrosinases may also increase digestibility of wood during decomposition by fungi due to their ligninolytic activity[61].

Hemocyanins represent a novel class of lignin-modifying proteins that have a pronounced impact on wood digestibility and likely underpin the unusual ability of *Limnoria* to live on a diet of wood in the absence of microbial assistance in digestion. Insights gained from this digestive system may prove useful in developing effective methods for producing sugars from lignocellulose for sustainable biofuel production. The opportunity to exploit seawater rather than valuable and limited freshwater resources may provide a further significant benefit.

## Methods

**Animal source**. The animal experiments described in this paper comply with and were approved by the Animal Welfare and Ethical Review Body of the University of Portsmouth (approval number 815 C). Specimens of *Limnoria quadripunctata* Holthuis were collected from a heavily infested piece of balau wood (*Shorea* sp.)

removed from a site in the intertidal zone at Portsmouth, UK. Specimens of the closely related *Limnoria tripunctata* Menzies were collected from heavily infested unidentified wood removed from the intertidal zone around the Isle of Wight, UK. Both species were used to set up laboratory cultures in seawater tanks.

**Mass balance experiment**. Ten batches of 10 animals (*L. quadripunctata*) were each fed for 28 days on a stick of willow wood or Scots pine sapwood (20 × 5 × 3 mm) previously leached for 1 week in water, dried for 48 h at 105 °C and accurately weighed. Fecal pellets produced during the experiment were collected from each batch by filtration. After the feeding period, animals were flash frozen, and the fecal pellets and remaining wood were washed and dried at 105 °C for 48 h before being weighed. Mass loss was calculated by subtracting the sum of weights of fecal pellets plus the remaining wood, from which the pellets had been generated, from the original weight of the wood before feeding.

**Biomass fraction analysis**. For biomass analysis the same ten samples of willow wood, Scots pine sapwood and fecal pellets (of 10 animals each) from the mass balance experiment were used. One aliquot (3–4 mg accurately weighed) milled wood and fecal pellets per sample was used for the determination of acetyl bromide-soluble lignin (ABSL)[63]; a second aliquot of each sample was used for the sequential sugar composition analysis of the matrix and crystalline polysaccharides using TFA and $H_2SO_4$. For lignin extraction, 250 μL of acetyl bromide solution (25% v/v acetyl bromide in glacial acetic acid) were added to the biomass and heated at 50 °C for 3 h mixing every 15 min. Cooled samples were transferred to 5 mL volumetric flasks, 1 mL of 2 M sodium hydroxide used to rinse the original tubes and 175 μL of 0.5 M hydroxylamine HCl added, and flasks filled up to 5 mL with glacial acetic acid. Before measuring absorbance at 280 nm spectrophotometrically, 100 μL of each sample were diluted with 900 μL glacial acetic acid. The amount of lignin was calculated using the following formula: % ABSL = (absorbance/(coefficient×path length)) × ((total volume × 100%)/biomass weight)) × dilution, where coefficient = 18.21 (poplar)[64]. For sequential extraction of monosaccharides from matrix and crystalline polysaccharide fractions, samples were washed with absolute ethanol and dried at 35 °C. Standards comprised a mixture of nine monosaccharides (arabinose, fucose, galactose, galacturonic acid, glucose, glucuronic acid, mannose, rhamnose, and xylose) each at 100 μM and were dispensed at 250, 500, and 700 μL into tubes in duplicates before drying in a speed vacuum concentrator (SPD131DDA, Thermo Scientific), used for all drying steps mentioned hereafter. One set of standards was hydrolyzed with TFA and the other with $H_2SO_4$ in the same way as the biomass samples. All samples were partially hydrolyzed by adding 0.5 mL of 2 M TFA to extract matrix polysaccharides first. Samples were flushed with dry argon and heated at 100 °C for 4 h with periodical mixing. Cooled samples were dried overnight, the pellets washed twice with 500 μL of 2-propanol and dried before TFA-soluble sugars were removed in two extractions with 500 μL of $dH_2O$ each at 35 °C. The combined TFA-extracts were dried along with the pellets, which were then used in the second hydrolysis step to extract crystalline polysaccharides. TFA-pellets were totally dissolved in 50 μL of 72% (w/w) $H_2SO_4$ and incubated at room temperature for 4 h, mixing every 15 min, then 1.05 mL of $dH_2O$ was added to reduce the $H_2SO_4$ concentration to 3.42% and incubated at 120 °C for 4 h. After cooling 1 μL of 1% bromophenol blue was added, samples were centrifuged and a 550 μL aliquot partially neutralized by adding 500 μL of 150 mM $Ba(OH)_2$. Complete neutralization was achieved by adding $BaCO_3$ powder until the solution turned blue, before samples were centrifuged to eliminate the precipitated $BaSO_4$ and supernatants were dried to be used as $H_2SO_4$-extracts for the quantification of monosaccharides. All dried TFA- and $H_2SO_4$-extracts of samples and standards were re-suspended in 200 μL of $dH_2O$, filtered with 0.45 μm PTFE filters, and analyzed by High-Performance Anion-Exchange Chromatography (HPAEC Dionex) using the monosaccharide program (see below). The amounts of all three biomass fractions analyzed (lignin, matrix and crystalline polysaccharides) were added together and the percentage of each fraction calculated from the total. The total mass of fecal pellets from all three fractions was set to 78% to represent the 22% mass loss during digestion.

**High-performance anion-exchange chromatography (HPAEC)**. Mono- and oligosaccharides were analyzed via HPAEC using an ICS-3000 PAD system with an electrochemical gold electrode, a CarboPac PA20 3 × 150 mm analytical column and a CarboPac PA20 3 × 30 mm guard column, operating with Chromeleon 6.8 Chromatography Data Systems software (Dionex). Sample and calibration standard aliquots of 5 μL were injected and separated at a flow rate of 0.4–0.5 mL min⁻¹ at a constant temperature. Parameters for the monosaccharide separation were as follows: after equilibration of the column with 100% $H_2O$, samples were injected and separated at a temperature of 25 °C in a linear gradient of 100% $H_2O$ to 99%–1% of $H_2O$–0.2 M NaOH in 5 min, then constant for 10 min, followed by a linear gradient to 47.5%–22.5%–30% of $H_2O$–0.2 M NaOH–0.5 M NaOAc/0.1 M NaOH in 7 min and then kept constant for 15 min. After washing the column with 0.2 M NaOH for 8 min it was re-equilibrated with 100% $H_2O$ for 10 min before injection of the next sample. Parameters for the oligosaccharide separation at 30 °C were as follows: after equilibration of the column with 50%-50% of $H_2O$–0.2 M NaOH, a linear gradient was started from 0 to 20% with 0.5 M NaOAc/0.1 M NaOH over 40 min and then kept constant for 6 min before being reverted to 50%–50% of $H_2O$–0.2 M NaOH

over 4 min before the next sample injection. Carbohydrates were identified by comparison with retention times of external standards and quantified by comparing integrated peak areas of samples to those of monosaccharide calibration standards, prepared as described above, and to oligosaccharide standards (glucose, cellobiose, cellotriose, cellotetraose, cellopentaose, cellohexaose) each at 125, 250, and 350 μM.

**Solid-state $^{13}C$ NMR on willow and fecal pellets**. Specimens of *L. tripunctata* were kept in seawater on willow sticks for 2 weeks and fecal pellets were collected daily by centrifugation and then kept in freshwater with 0.02% Na-azide at 4 °C. Control wood was milled using a cyclone mill (Retsch) with a 0.5 mm mesh and soaked in seawater before being transferred into $H_2O$/azide and kept in the same way as the fecal pellets. Magic Angle Spinning (MAS) ssNMR experiments on wood and fecal pellets were performed on a Bruker Advance III NMR spectrometer operating with TopSpin software version 3.5 at $^{1}H$ and $^{13}C$ Larmor frequencies of 398.8 and 100.3 MHz, respectively, using a 4.0 mm double-resonance MAS probe. Experiments were conducted at room temperature and a MAS frequency of 12 kHz. The $^{13}C$ chemical shift was determined using the carbonyl peak at 177.8 ppm of L-alanine as an external reference with respect to tetramethylsilane; the $\pi/2$ pulse lengths were 2.5 μs ($^{1}H$) and 4.2 μs ($^{13}C$). $^{1}H$-$^{13}C$ cross-polarization (CP) with ramped (70–100%) $^{1}H$ RF amplitude was used to obtain the spectra with 100 kHz $^{1}H$ decoupling during acquisition and a contact time of 1 ms. The recycle delay was 5 s with two blocks of 12,000 acquisitions added for each sample.

**ATR-FTIR spectroscopy**. Fecal pellets from balau wood colonized by *L. quadripunctata* were collected by centrifugation and dried, uncolonized parts of the same wood were shaved off and dried before being milled to fine powder using a ball mill with three cycles of 5 min milling each. ATR-FTIR spectra of wood powders and fecal pellets were obtained between 850–1850 cm⁻¹ using a Spectrum One spectrometer equipped with a diamond that allows collection of spectra directly on the sample without any sample preparation (Perkin-Elmer). Three spectra for each sample were acquired with Spectrum version 5.0.1 using 256 scans at a resolution of 4 cm⁻¹, and the triplicate-averaged spectrum was used for principal component analysis (PCA) of three wood and fecal pellet samples, respectively. PCA was carried out using The Unscrambler X software v 10.5 (CAMO) after peak normalization, linear baseline correction, and area normalization. Spectral assignments were made according to the literature (see Supplementary Table 2).

**Scanning electron microscopy (SEM)**. Fecal pellets were obtained from a laboratory culture of *L. quadripunctata* and hindguts were removed by dissection and promptly incubated in fixative for 1 h. Fecal pellets were fixed in 4% glutaraldehyde in 0.2 M sodium cacodylate and 2 mM calcium chloride at pH 7.4. Hindguts in 0.1 M sodium cacodylate, 3% paraformaldehyde and 0.5% glutaraldehyde pH 7.4. Tissues were rinsed in 0.2 M sodium cacodylate pH 7.4 before post-fixation with 1% (w/v) aqueous osmium tetroxide for 1 h. After 3 × 30 min rinses in Reverse Osmosis water, the samples were dehydrated through a graded ethanol series, into acetone and transferred into hexamethyldisilazane, then dried by evaporation. They were placed onto adhesive carbon tabs and sputter coated with gold. SEM images were obtained with a JEOL 6060 LV microscope operating at 15 kV in secondary electron mode.

**Transmission electron microscopy (TEM)**. Hepatopancreases were removed from *L. quadripunctata* by dissection and fixed and dehydrated as described for fecal pellets used for SEM (s.a.) before infiltration with low viscosity resin (Agar Scientific) and polymerization at 60 °C for 16 h. For TEM, gold sections (70 nm) were cut using a Leica Ultracut UCT and mounted on 200 mesh copper grids. Sections were post-stained with 2% aqueous uranyl acetate (10 min) and lead citrate (5 min in a carbon dioxide-depleted chamber). Sections were viewed with an FEI Technai G2 TEM operating at 120 kV.

**Measurement of pH values within the digestive tract**. Animals (*L. quadripunctata*) were kept in sticks of pine wood (*Pinus sylvestris*) (20 × 2 × 4 mm) at 18–20 °C in seawater (taken from Langstone Harbour, Portsmouth, UK) for a month. At the day of the experiment, the animals were removed from the sticks and fixed by the cephalon and the telson with acupuncture needles to agarose plates (2% in Langstone Harbour seawater). The thoracic segments were removed with the help of tweezers to expose the hindgut and then the plate was filled up with seawater. Immediately afterwards, a 25 μm-tip pH microsensor (Unisense, Denmark) was introduced from above into the hindgut lumen. A pH profile was set up to measure the pH every 25 μm, with 3 s of acquisition and 3 s of rest between measurements, starting inside the hindgut lumen going onwards into the agarose plate, with a motorized micromanipulator (MM33 Unisense, Denmark). The profiles were acquired using the SensorTrace Suite software (Unisense, Denmark). Hepatopancreases were dissected and embedded in 12% gelatin then placed on top of agarose plates. A 25 μm tip pH microsensor (Unisense, Denmark) was used to measure the pH in the four hepatopancreas lobes. Measurements were made in the proximal region of the small and large lobes, near the connection to the hindgut, and in a more distal region in the larger lobes. The sensor tip was introduced into the tissue and the pH was recorded after the signal was stable.

Measurements of pH were compared by Welch's one-way analysis of variance with region of measurement as the factor (distal region of large lobe of hepatopancreas, proximal region of large lobe, proximal region of small lobe, hindgut, seawater/agarose support for hindgut). Comparisons were made without assuming equality of variances using the Games-Howell multiple comparison method. Comparisons between the measurements in various regions in the hepatopancreas lumen were made with Tukey's multiple comparison method applied to a generalized linear model model using factors of region of measurement, left or right and individual animal. Tests were conducted using Minitab version 17.3.1.

**Measurement of oxygen levels within the digestive tract**. Oxygen levels were measured in situ in the hindgut of live individuals of *L. quadripunctata* mounted in a Petridish on a layer of agar covered with seawater. Individuals were placed dorsally on the agar surface and suspended in a cool layer of 12.5% (w/v) aqueous gelatin. The oxygen microelectrode was calibrated against a mixture of 0.1 M ascorbic acid and 0.1 M sodium hydroxide to provide a depleted oxygen standard, as well as against air-saturated $dH_2O$, which had been vigorously bubbled for 5 min. Using a micromanipulator, the 10 μm-tip of the microelectrode was passed through the cuticle of the ventral surface into the hindgut and then stepwise withdrawn with 3 s each of data acquisition and of rest between measurements at 21 °C. After removal of the electrode and release, the live specimens were observed to swim and feed normally.

**Transcriptomics of the digestive system and whole animals**. Three times, 25 whole animals (*L. tripunctata*) were collected in seawater on ice, liquid removed and flash frozen in liquid nitrogen before homogenization with a micropestle and total RNA extraction with the TRIzol® Reagent (Thermo Fisher Scientific). Triplicate biological samples were prepared from hindgut and hepatopancreas by dissection of 50 animals per sample, which were collected in TRIzol® Reagent (Thermo Fisher Scientific) on ice, ground with a micropestle and total RNA extracted according to the manufacturer's protocol. Total RNA was treated with RQ1 DNase (Promega) to remove genomic DNA, cleaned with RNA Clean & Concentrator™-5 (Zymo Research), then quantified spectrophotometrically (NanoDrop 2000, Thermo Scientific) and its integrity evaluated using the Bioanalyzer2100 (Agilent). The transcriptomic library was constructed after poly-A selection using the TruSeq® Stranded mRNA Sample Preparation Kit (Illumina) according to the library protocol. The 150 bp paired end sequencing was performed with HiSeq3000 using Illumina Technology (The Next Generation Sequencing Facility, University of Leeds). A total of 377,501,758 raw EST read pairs (PRJNA453115; SRP142516) were trimmed to remove adaptors, cleaned of low-quality reads and then assembled into contigs (represented with 478,804,012 total assembled bases in 1,062,392 Trinity transcripts) with an average contig length of 451 bp using the Trinity software v2.5.1[65]. Raw reads from each library were mapped onto this assembly and mapped reads were counted with SAMtools. The final transcriptome of 767,816 sequences was generated after filtering out contigs with fewer than five reads mapped in all libraries and used as a reference database for Mascot searches in our proteomic study.

Annotation of the contigs identified in our proteomic study was performed by BlastX searches against the non-redundant database of NCBI. Additional CAZyme annotation was carried out using the online software dbCAN (DataBase for automated Carbohydrate-active enzyme ANnotation) after the contigs were converted into ORFs using the online tool Emboss (http://www.bioinformatics.nl/cgi-bin/emboss/getorf).

**Label-free quantitative proteomic analyses**. For digestome analysis triplicate biological samples were prepared from hindgut and hepatopancreas by dissection of 100 animals (*L. tripunctata*) per sample and separation of tissue and solid content from the digestive fluids by mild centrifugation at 3000×*g* for 5 min in 0.04 M NaPO₄ pH 7, containing 1× Halt protease inhibitor cocktail ethylene diamine tetra-acetic acid (EDTA)-free (Pierce) and 0.01% TritonX100. Fluid samples were dried in a speed vacuum concentrator (SPD131DDA, Thermo Scientific) and tissue/solid samples were ground with a micropestle before being solubilized in NuPAGE LDS sample buffer (Life Technologies) with heating at 70°C for 10 min and running into a 7 cm NuPAGE Novex 10% Bis-Tris gel (Life Technologies) at 200 V for 6 min. For analysis of purified hemocyanin, native hemocyanin extract was mixed with sample loading buffer and treated as described above. Gels were stained with SafeBLUE protein stain (NBS biologicals) for a minimum of 1 h before de-staining with ultrapure water for a minimum of 1 h.

In-gel tryptic digestion of proteins was performed after reduction with dithioerythritol and S-carbamidomethylation with iodoacetamide. Gel pieces were washed two times with aqueous 50% (v/v) acetonitrile containing 25 mM ammonium bicarbonate, then once with acetonitrile and dried in a vacuum concentrator for 20 min. Sequencing-grade, modified porcine trypsin (Promega) was dissolved in 50 mM acetic acid, then diluted fivefold with 25 mM ammonium bicarbonate to give a final trypsin concentration of 0.02 μg μL⁻¹. Gel pieces were rehydrated by adding 25 μL of trypsin solution, and after 10 min enough 25 mM ammonium bicarbonate solution was added to cover the gel pieces. Digests were incubated overnight at 37°C before extraction of peptides by washing three times with aqueous 50% (v/v) acetonitrile containing 0.1% (v/v) trifluoroacetic acid, and

drying in a vacuum concentrator and reconstituting in aqueous 0.1% (v/v) trifluoroacetic acid. A common sample pool was created by taking equal aliquots of all samples.

For LC-MS/MS analysis, samples were loaded onto an UltiMate 3000 RSLCnano HPLC system (Thermo) equipped with a PepMap 100 Å $C_{18}$, 5 μm trap column (300 μm×5 mm, Thermo) and a PepMap, 2 μm, 100 Å, $C_{18}$ EasyNano nanocapillary column (75 μm×500 mm, Thermo). The trap wash solvent was aqueous 0.05% (v/v) trifluoroacetic acid and the trapping flow rate was 15 μL min⁻¹. The trap was washed for 3 min before switching flow to the capillary column. Separation used gradient elution of two solvents: solvent A, aqueous 1% (v/v) formic acid; solvent B, aqueous 80% (v/v) acetonitrile containing 1% (v/v) formic acid. The flow rate for the capillary column was 300 nL min⁻¹ and the column temperature was 40 °C. Analyses were performed over 1 h (hemocyanin extracts) or 3 h (tissue and fluid extracts) acquisitions. For 1 h runs the linear multi-step gradient profile was: 3–10% B over 7 min, 10–35% B over 30 min then 35–99% B over 5 min. For 3 h acquisitions the gradient profile was: 3–10% B over 7 min, 10–35% B over 30 min, 35–99% B over 5 min and in both cases then proceeded to wash with 99% solvent B for 4 min. The column was returned to initial conditions and re-equilibrated for 15 min before subsequent injections. The nanoLC system was interfaced with an Orbitrap Fusion hybrid mass spectrometer (Thermo) with an EasyNano ionization source (Thermo). Positive ESI-MS and MS2 spectra were acquired using Xcalibur software (version 4.0, Thermo). Instrument source settings were: ion spray voltage, 1,900 V; sweep gas, 0 Arb; ion transfer tube temperature; 275 °C. MS1 spectra were acquired in the Orbitrap with: 120,000 resolution, scan range: *m/z* 375–1500; AGC target, 4e⁵; max fill time, 100 ms. Data dependent acquisition were performed in top speed mode using a fixed 1 s cycle, selecting the most intense precursors with charge states 2–5. Easy-IC was used for internal calibration. Dynamic exclusion was performed for 50 s post precursor selection and a minimum threshold for fragmentation was set at 5e³. MS2 spectra were acquired in the linear ion trap with: scan rate, turbo; quadrupole isolation, 1.6 *m/z*; activation type, HCD; activation energy: 32%; AGC target, 5e³; first mass, 110 *m/z*; max fill time, 100 ms. Acquisitions were arranged by Xcalibur to inject ions for all available parallelizable time.

For the data analysis, peak lists were converted from.raw to.mgf format using MSConvert (ProteoWizard 3.0.9974) before submitting to a locally-running copy of the Mascot program using Mascot Daemon (version 2.5.1, Matrix Science). Data were searched against an in-house *Limnoria* transcriptomic database (see above; PRJNA453115; SRP142516) with the following criteria specified: Enzyme, trypsin; Max missed cleavages, 2; Fixed modifications, Carbamidomethyl (C); Variable modifications, Oxidation (M), Peptide tolerance, 3 ppm; MS/MS tolerance, 0.5 Da; Instrument, ESI-TRAP. Search results were passed through Mascot Percolator to achieve a 1% peptide false discovery rate and filtered to require a minimum expect score of 0.05 for individual matches. Protein identifications were filtered to require a minimum of two peptide matches in the purified hemocyanin extract, and at least in one sample when comparing organ-specific abundances, with samples being triplicates each of gut tissue/solids, gut fluids, hepatopancreas tissue/solids, and hepatopancreas fluids. Relative protein molar abundances were calculated from Mascot derived emPAI scores as described by Ishihama et al.[66].

**Protein ID**. The digestion was the same as for LC-MS/MS workflow (see above, Label-free quantitative proteomic analysis), but protein ID used MALDI-MS/MS for analysis. A 1 μL aliquot of each peptide mixture was applied directly to the ground steel MALDI target plate, followed immediately by an equal volume of a freshly-prepared 5 mg mL⁻¹ solution of 4-hydroxy-α-cyano-cinnamic acid (Sigma) in 50% aqueous (v/v) acetonitrile containing 0.1% trifluoroacetic acid (v/v). Positive-ion MALDI mass spectra were obtained using a Bruker ultraflex III in reflectron mode, equipped with a Nd:YAG smart beam laser. MS spectra were acquired over a mass range of *m/z* 800–4000. Final mass spectra were externally calibrated against an adjacent spot containing six peptides of known mass. For each spot the 10 most intense ions, with S/N > 30, were selected for fragmentation, which was performed in LIFT mode without the introduction of a collision gas. The default calibration was used for MS/MS spectra, which were baseline-subtracted and smoothed (Savitsky-Golay, width 0.15 m/z, cycles 4). Bruker flex-Analysis software (version 3.3) was used for spectral processing and peak list generation. Monoisotopic masses were obtained using a SNAP averaging algorithm (C 4.9384, N 1.3577, O 1.4773, S 0.0417, H 7.7583) and a S/N threshold of 2 for MS and 6 for MS2. Tandem mass spectral data searched against an in-house *Limnoria* database (73,986 sequences; 9,177,896 residues)[21] using a locally-running copy of the Mascot program (Matrix Science Ltd., version 2.4), through the Bruker ProteinScape interface (version 2.1). Search criteria specified: Enzyme, Trypsin; Fixed modifications, Carbamidomethyl (C); Variable modifications, Oxidation (M); Peptide tolerance, 100 ppm; MS/MS tolerance, 0.5 Da; Instrument, MALDI-TOF-TOF. Peptide matches were filtered to require an expect score of 0.05 or lower.

**Gene expression analysis**. Three sets of five *L. quadripunctata* (for hemocyanin Hc) or one set of 50 animals (for GHs) each were dissected to separate their hindgut, hepatopancreas, and the rest of the body, which were placed immediately into TRIzol® Reagent (Thermo Fisher Scientific) to extract total RNA according to the manufacturer's protocol. The RNA concentration was determined spectro-photometrically (NanoDrop 2000, Thermo Scientific) and its integrity evaluated

using the Bioanalyzer (Agilent). Total RNA (400 [GHs]–500 [Hc] ng) was treated with RQ1 DNase (Promega) to remove genomic DNA, and subjected to reverse transcription (RT) using oligo dT primers and SuperScriptII RT (Life Technologies) according to the manufacturer's instructions. Quantitative polymerase chain reaction (qPCR) of diluted cDNA and no-RT controls (equivalent to 10 ng total RNA per reaction) was performed with Fast SYBR® Green Master Mix according to the manufacturer's protocol on a StepOnePlus™ Real-Time PCR System (Life Technologies) as technical triplicates. Specific primers for *L. quadripunctata* genes coding for hemocyanin (GenBank accessions: GU166295 [*Hc1*], GU166296 [*Hc2*], GU166297 [*Hc3*], GU166298 [*Hc4*]), GH family 5, 7 and 9 (GenBank accessions: GU066826 [*GH5A*], GU066827 [*GH5C*], FJ940756 [*GH7A*], FJ940757 [*GH7B*], FJ940759 [*GH9A*], FJ940760 [*GH9B*]), ubiquitin (*Ubi*), and glyceraldehyde 3-phosphate dehydrogenase (*GAPDH*) were used: *Hc1*, TCTGCTATTGTTTCAC GACTTAATCAT, TAGCGAAGACATCTGCTGCATT; *Hc2*, CACATTACCAG GAAATCAAAGGATACT, ACGGCAGCGTCAGCTTGT; *Hc3*, GATCTAACCCA CATAACACGAAAATCAT, GAATAAGCAGACAAGTC CAAATCC; *Hc4*, AACCACGTAACCCGAAAATCAT, TGAATAAGCAGACAAGTC CAAATCC; *GH5A*, CGGTATGGTAACTTCTTGCGATAGTAG, GTCGCTT TGCTGGGCACTA; *GH5C*, GCTTCAATCTTACCTTGATAACTGTTTG, CAT GCTCCTAAAGCTGGATGGT; *GH7A*, CCAAATGCAGGAACTGGTGAT, GC CATGCTATTAGCTTCCCAAATA; *GH7B*, TTGCTGGCAAAGCTAATTCTG AT, GCAGCAGGCGTCCCATTTGT; *GH9A*, TGCATCAGCTCCAGGTACTGA T, GCGTATGAACCCGAATGGA; *GH9B*, CACCAAACATCCTACGTCAACA GT, CCCCAGCCTAATTCATCTTGAA; *Ubi*, GGTTGATCTTTGCCGGAAAG, TCTCAAAACGAGGTGAAGTGTTG; *GAPDH*1, TGTAATTTTCCTTCCATC GACAAC, CTCCACACACGGTCGCTACA; *GAPDH*2, CTCTACCTCCGCGC CAATC, CGCTGTAACGGCTACTCAGAAGA. Quality controls have been employed according to the MIQE-guidelines[67] and included verification of primer specificity, testing for contamination of RNA with genomic DNA and of the PCR mix and evaluation of consistency of cDNA synthesis, all as previously described[22]. Gene expression levels are shown as previously described relative ubiquitin-normalized efficiency-corrected values[22].

**SDS–PAGE and western blot analysis.** For tissue-specific immunoblot detection 50 *L. quadripunctata* specimens were dissected to remove hepatopancreases and hindguts from which proteins were extracted by grinding material in 50 mM Tris-HCl pH 8, 300 mM NaCl, 0.1% Tween 20, 1× Halt protease inhibitor cocktail (Pierce) before pelleting any cell debris by centrifugation. The protein concentration was estimated using Bradford reagent (Pierce). After separation of 5 µg protein (organ and solid content) alongside a molecular weight marker (PageRuler Plus Prestained Protein Ladder, Thermo Fisher Scientific) on polyacrylamide gels using standard denaturing discontinuous SDS–PAGE, proteins were either visualized by Coomassie stain with InstantBlue (Expedeon) or transferred onto Protran85 nitrocellulose membranes (Whatman) by semidry blotting in Towbin buffer for western analysis. Fractions from gel filtration of native *Limnoria* extracts were separated on SDS–PAGE gels and blotted as described above. Primary polyclonal antibodies for detection of *Limnoria* epitopes of hemocyanin (used at 1:1000 dilution) and GH9 (used at 1:500 dilution) were raised in rabbit and for GH5 (used at 1:500 dilution) and GH7 (used at 1:5000 dilution) in sheep (see below) and detected by stabilized goat anti-rabbit IgG (H + L) secondary antibody, horseradish peroxidase (HRP) (Invitrogen, 32460; used at 1:1250 dilution) or by secondary rabbit anti-sheep IgG antibody, HRP conjugate (Sigma, AP147P; used at 1:10,000 dilution), respectively. SuperSignal West Pico Chemiluminescent Substrate (Pierce) was added before exposure to ECL Hyperfilm (GE Healthcare) to visualize protein accumulation. All uncropped gel and blot images are shown in Supplementary Fig. 12.

**Antibody production and purification.** The sequences encoding LqGH5A (GU066826), LqGH7A (FJ940756), and LqGH9A (FJ940759) lacking their signal peptide were inserted into the NheI/EcoRI sites of the vector pET28A as a C-terminal 6×His-tagged fusion. The resulting constructs were transformed into BL21 *Escherichia coli* strain and fresh colonies were picked from Lysogeny broth plates containing chloramphenicol 34 mg L$^{-1}$ and kanamycin 50 mg L$^{-1}$. A 5 ml culture of three colonies per construct was set up in the same medium and grown overnight at 37 °C under agitation. Two mL of these pre-cultures were used to inoculate 200 mL of 2YT medium and cultures were grown to early exponential phase (OD600 = 0.5–0.6), induced with 1 mM isopropyl β-D-1-thiogalactopyranoside and then grown overnight at 30 °C. Cells were harvested by spinning cultures down at 3000 g for 5 min and re-suspended in 100 mM NaH$_2$PO$_4$, 10 mM Tris-Cl, 8 M urea following manufacturer instructions (Qiagen). Preparation of clarified cell extracts and purifications of LqGH5A, 7 A and 9 A under denaturing conditions were performed using a Nickel-NTA column (Qiagen). Preparation purity was assessed by SDS–PAGE and sample concentration was determined by Bradford (Pierce). About 1 mg of recombinant proteins LqGH5A and 7 A was used to raise polyclonal antibodies in sheep (Scottish National Blood Transfusion Service, Penicuik, UK) and ~ 500 µg of recombinant LqGH9A were used to raise polyclonal antibodies in rabbit (Covalab S.A.S., France).

The sequence encoding LqHc1 (GU166295) lacking the signal peptide was inserted into the AgeI/KpnI sites of the vector pHLsec by In-Fusion cloning (Clontech) as a C-terminal His$_6$-tagged fusion[68]. The resulting construct was

transiently transfected into an adherent human embryonic kidney cell culture (HEK-293T) using purified plasmid DNA and lipofectamine 2000 (Life Technologies). Prior to transfection, cells were grown at 37 °C, 5% CO$_2$, 90% RH in Dulbecco's modification of Eagle medium containing 10% fetal bovine serum (FBS) and supplemented with L-glutamine and non-essential amino acids to achieve 90–95% confluency. For transfection, the medium was exchanged to contain 1.5% FBS and cells were further incubated as described above. After 3 days cells were harvested by pelleting at 700 g for 5 min at 4 °C and re-suspended in denaturing lysis buffer (50 mM Tris-Cl pH 8, 150 mM NaCl, 8 M GuHCl) before the addition of DTT to 1 mM and sonication. For His-purification, the lysate was incubated sequentially with two aliquots of Ni-NTA agarose (Qiagen) and purified according to The QIAexpressionist protocol using denaturing buffer B containing 20 mM imidazole for washing and 500 mM imidazole for elution of recombinant proteins. The purified LqHc1 was dialyzed against PBS and ~ 200 µg of protein was used to raise polyclonal anti-hemocyanin antibodies in rabbits (Covalab S.A.S., France).

To enrich the sera for antigen-specific antibodies, affinity columns were made using recombinant LqGH5A and LqGH9A purified as described above, LqGH7B purified from *Aspergillus oryzae* culture supernatants as described in Methods Expression and Purification[22] or using recombinant His-tagged LqHc3 (GU166297) purified from a bacterial culture under denaturing conditions on Ni-NTA resin using decreasing pH for wash and elution steps (The QIAexpressionist). Recombinant protein preparations were dialyzed against coupling buffer (0.1 M NaHCO$_3$, 0.5 M NaCl, pH 8.3) and bound to CNBr-activated Sepharose™ 4 Fast Flow resin (GE Healthcare Life Sciences) followed by affinity purification of an aliquot of each crude antibody serum according to the resin manufacturer's instructions. Purified antibody fractions were characterized by their affinity to each Lq antigen by dot and western blot using both recombinant Lq protein and *L. quadripunctata* whole body extracts. Fractions showing the highest titer and no unspecific binding were selected for western blot experiments.

**Purification of native *Limnoria* hemocyanin.** Fifty animals (*L.* spp.) were homogenized in 0.05 M NaPO$_4$ buffer pH 7, containing 1× Halt protease inhibitor cocktail EDTA-free (Pierce), centrifuged 5 min at 17,000 g and filtered (0.45 µm). The (concentrated) extract was analyzed by gel filtration chromatography using the ÄKTApurifier UPC10 with UNICORN 5.31 workstation and a Superose 6 Increase 10/300 GL column (GE healthcare) pre-equilibrated with filter-sterilized seawater or 0.05 M NaPO$_4$ pH 7 at a flow rate of 0.2–0.4 mL min$^{-1}$, and fractions (0.5 mL) were collected after 0.3–1 CV. Fractions of peak 1 and 2 containing hemocyanin were pooled and concentrated before protein concentration was estimated using Bradford reagent (Pierce) or before proteins were subjected to LC-MS/MS analysis.

**Uranyl acetate negative stain.** A few µL of native *Limnoria* hemocyanin extract were left on a copper grid (with formvar/carbon support film) for 4 min then negative stained with 1% uranyl acetate in water before being imaged with a FEI Tecnai 12 TEM, operating at 120 kV, and using a Ceta camera.

**UV-Vis spectroscopy.** Native *Limnoria* hemocyanin (160 µg mL$^{-1}$, 2.2 µM) or *Trametes versicolor* laccase (184 µg mL$^{-1}$, 3.3 µM, equivalent to 1U; Sigma) were incubated with 1 mM pyrogallol in 100 µL of 0.1 M NaPO$_4$ pH 6.8 at room temperature for up to 45 min. Enzyme activity of hemocyanin was induced with 2.75 mM SDS and buffer controls only lacked the protein. Ultra violet-Visible absorbance spectra of aliquots were scanned from 220 to 750 nm in a NanoDrop 8000 Microvolume UV-Vis spectrophotometer (Thermo Scientific) at regular intervals.

To monitor the effect of chelators on oxygen binding to the copper center of hemocyanin, 50 mM of either ethylene diamine tetra-acetic acid (EDTA), ethylene glycol-bis(2-aminoethylether)-N,N,N′,N′ tetra-acetic acid (EGTA), or diethylene triamine penta-acetic acid (DTPA) was mixed with hemocyanin extract in seawater and scanned as above.

**Solid-state $^{13}$C NMR spectroscopy of lignin.** After incubation of 3% (w/v) alkali lignin (Sigma #471003 low sulfonate content, 3.3% sulfur and 50.1% carbon) in seawater supplemented with 0.18 mM FeSO$_4$ and with or without 50 mM DTPA with native *Limnoria* hemocyanin (~ 14 µM) or seawater (control) in triplicates each for 1 h at room temperature, reactions were freeze-dried before analysis by ssNMR. Experiments were performed using a Bruker Avance 400 spectrometer, equipped with a Bruker 4.0 mm double-resonance MAS probe, at $^{13}$C and $^1$H frequencies of 100.5 and 400.0 MHz, respectively. The spinning frequency at 14 kHz was controlled by a pneumatic system that ensures rotation stability higher than ~ 1 Hz. Typical π/2 pulse lengths of 4 and 3.5 µs were applied for $^{13}$C and $^1$H, respectively. $^1$H decoupling field strength of γB1/2π = 100 kHz was used. $^{13}$C Cross-Polarization (CP)–MAS NMR spectra was measured using Multi-CP excitation[69,70] with nine cross-polarization blocks of 1 ms and one last cross-polarization of 0.8 ms, 90–100% increment in the RF amplitude, repolarization period $t_z$ of 0.9 s, and recycle delay of 2 s. The same experimental conditions were used in all experiments. Bruker TopSpin software version 3.5 was used for data collection, and OriginLab OriginPro 9.0.0 SR2 for data analysis. Chemical shifts were assigned based on literature data (Supplementary Note 1).

**Thermal shift assays (thermofluor)**. Thermal shift assays were conducted on purified proteins with SYPRO™ Orange Protein Gel Stain (Life Technologies) using an Mx3005P qPCR System (Agilent Technologies). The intensity of the fluorescence was measured against a temperature gradient of 25–95 °C and values plotted to determine the melting temperature ($T_m$) by curve fitting using a five parameter sigmoid equation with the $T_m$ measured as the midpoint at http://paulsbond.co.uk/jtsa[71].

**Digestibility assays**. Willow wood and Scots pine sapwood was ground using a cyclone mill (Retsch) with a 1 mm mesh and a ball mill with three cycles of 5 min milling each and then 10 mg were aliquoted into 2 mL screw cap tubes. After shaking incubation of wood powder (10% w/v) or phosphoric acid swollen cellulose (PASC; 4% w/v) in seawater supplemented with or without 50 mM DTPA, or 50 mM sodium dithionite, with native *Limnoria* hemocyanin (~14 μM), horse spleen ferritin (Sigma; 1.9 μM) or seawater alone (control) for 10–20 min at room temperature, the solids were pelleted and pretreatment solutions removed. For pretreatments of wood powder with 14 μM *Trametes versicolor* laccase (Sigma) or *Agaricus bisporus* mushroom tyrosinase (Sigma) the reactions were incubated and processed as above but in 0.05 M NaPO$_4$ at pH 6. For hemocyanin re-use experiments, the hemocyanin- or seawater-containing pretreatment solutions were removed from replicates after pelleting the solids, then pooled and re-aliquoted onto new willow wood samples in equal volumes for the new pretreatment reaction; as some of the liquid had been taken up by the wood the total volume applied in the re-use experiment was slightly less than in the previous one (86 μL instead of 100 μL). Mild thermochemical pretreatments consisted of incubation of wood powder with 0.5 N NaOH for 45 min at 90 °C followed by extensive washing in buffer to adjust the pH. For saccharification reactions, the pretreated wood powders and PASC were incubated with 10 μg (10% w/v) recombinant cellobiohydrolase I from *Hypocrea jecorina* (*Hj*CBH I or Cel7A, Sigma) or *L. quadripunctata* cellobiohydrolase Cel7B (LqGH7B, Novozymes[22]) in 0.05 M NaPO$_4$ at pH 6–7, or with buffer only as no GH controls, for 5 h (wood) or 2 h (PASC), respectively, at 37–40 °C with shaking. Carbohydrate composition of the hydrolysis reaction was determined by HPAEC as described above after removal of solids by centrifugation, precipitation with 80% ethanol, drying of the supernatant containing mono- and oligosaccharides, resuspension in water and filtering through 0.2 μm polytetrafluoroethylene (PTFE) filters.

**Cellulase binding to hemocyanin-pretreated biomass**. Pretreatment and sachharification reactions were performed as described above for digestibility experiments, using 10 mg willow wood powder, native *Limnoria* hemocyanin (~14 μM) or seawater for 10 min in pretreatment reactions at room temperature, followed by 5 h saccharification with 10 μg (10% w/v) recombinant cellobiohydrolase I from *Hypocrea jecorina* (*Hj*CBH I or Cel7A, Sigma) in 100 μL 0.05 M NaPO$_4$ pH 6 at 37 °C. Aliquots 20 μL in SDS-loading buffer of hydrolysate (16 μL aliquoted from 100 μL hydrolysate after removal from biomass plus 4 μL of 5× SDS-loading buffer) or biomass (100 μL 1× SDS-loading buffer added to the biomass after pelleting and removal of hydrolysate) compared with the pretreatment and saccharification solutions prior use (16 μL plus 4 μL 5× SDS-loading buffer each) were loaded onto 4–20% Mini-PROTEAN ® TGX™ Precast Protein Gels (Bio-Rad) alongside a molecular weight marker (PageRuler Plus Prestained Protein Ladder, Thermo Fisher Scientific) after boiling at 100 °C for 5 min. Proteins were visualized by Coomassie stain with InstantBlue (Expedeon).

**NMR relaxometry of hemocyanin-treated wood**. Ten 10 mg willow samples were pretreated with either 100 μL seawater or hemocyanin (~14 μM) each for 3 h at room temperature and solutions removed after pelleting the biomass, which was then dried, pooled and analyzed according to the following methodology.

To assess the pore structure of the materials, we used *N,N*-Dimethylacetamide (DMAc, high-performance liquid chromatography grade) as a molecular probe. Although water is often used to determine pore structure in relaxometry experiments, it alters the pore distributions in biomass samples. As the interaction of DMAc and the cell wall is rather weak, the use of this molecule does not significantly change the pore structures[41]. The samples were dried in a vacuum oven at 510 mm Hg and 80 °C during 24 h. Then, they were soaked with DMAc that had been dried of water using molecular sieves (Sigma-Aldrich 3 Å, beads 4–8 mesh) and pumped at 570 mm Hg during 20 min to fill the sample pores, after which excess DMAc was removed with a micropipette.

Carr-Purcell-Meiboom-Gill NMR experiments (CPMG decays) were performed using a Bruker Minispec MQ-20 spectrometer operating with a magnetic field of 0.5 T ($^1$H Larmor frequency of 20 MHz). In total, 50,000 echoes were acquired with echo times of 70 μs and recycle delays of 15 s. Data were acquired using the Bruker minispec software v2.59 and measured in 32 repeated scans with signal-to-noise ratios of 199 (control sample) and 367 (hemocyanin-treated sample). The CPMG decay curves were processed in MathWorks MATLAB R2015a to obtain the $T_2$ distribution using a non-negative least square procedure also known as a numerical Inverse Laplace Transform, ILT[72,73], with the ILT MATLAB code from Schlumberger Doll Inc.[74]. The obtained $T_2$ distributions were deconvoluted using log-Gaussian functions to provide the contribution of each component in the pore structure. To make the deconvolutions, the distributions were normalized to have

unit area, and the logarithm of the $T_2$ axis of the distribution of each sample was calculated and then fitted, using the Curve Fitting Toolbox of MATLAB, with three Gaussian functions given by

$$f(x) = ae^{-(x-b)^2/2c^2} \qquad (1)$$

From the model, the relative areas are obtained from the relation

$$A = \sqrt{2\pi}ac \qquad (2)$$

and the error bars defined using error propagation from the standard deviations of the fitted parameters $a$ and $c$. The fluid enclosed in a pore interacts with the pore surface, restraining the molecular mobility of the fluid. As the surface-to-volume ratio $S/V$, defined as the inverse of the characteristic pore size, increases, the stronger is the interaction. The restriction on mobility is reflected by a decrease on the transverse relaxation time $T_2$ and can be quantified, on the fast diffusion regime, by the relation[75,76]

$$1/T_2 = \rho(S/V) = 2\rho/r \qquad (3)$$

where rho is the surface relaxivity and $S$ and $V$ are the pore surface area and the volume, respectively. The surface relaxivity constant depends on the particular porous material[77], with its surface properties such as magnetic susceptibility and affinity to the enclosed fluid defining its ability to produce spin relaxation of the DMAc protons. This parameter is usually unknown, turning the estimation of the actual pore sizes, or interstitial scales in this case, into a daunting challenge. In a realistic scenario, the distribution of pore sizes and differences in the fluid mobility within the pores result in a multi-exponential decay of the CPMG signal, i.e., a distribution of $T_2$ times. For DMAc in cotton cellulose $T_2$ relaxation times ranging from $10^{-4}$ to 1 s have been reported, corresponding to pore sizes of few nanometers to few micrometers[41]. Larger $T_2$ values on the distribution profile correspond to larger pores[77].

## Data availability

All transcriptomic raw sequence data were deposited in the NCBI BioProject database (PRJNA453115) and Sequence Read Archive (SRA) (SRP142516). All proteomic data sets, including raw data files, processed peak lists, and database search results have been deposited to the ProteomeXchange Consortium via the MassIVE partner repository with the dataset identifier PXD009486 and the MassIVE accession code MSV000082271. A reporting summary for this Article is available as a Supplementary Information file. All other data are available from the corresponding authors upon reasonable request.

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

### Acknowledgements

From the University of York we thank Jared Cartwright for advice on protein purification and Meg Stark for negative staining and imaging of native hemocyanin, both Bioscience Technology Facility, and Phil Roberts for design of the *Limnoria* schematic drawing. This work was funded by the Biotechnology and Biological Sciences Research Council (BBSRC), UK (Grants BB/G016178/1 and BB/L001926/1). I.P., G.P.C., J.G.F., and E.R.A. acknowledge funding from the São Paulo Research Foundation (FAPESP), Brazil (grants 2015/13684–0 and 2017/24465–3) and from the National Council for Scientific and Technological Development (CNPq), Brazil (grants 423693/2016–6 and 312852/2014–2). The York Centre of Excellence in Mass Spectrometry was created thanks to a major capital investment through Science City York, supported by Yorkshire Forward with funds from the Northern Way Initiative, and subsequent support from EPSRC (EP/K039660/1; EP/M028127/1).

### Author contributions

K.B., S.M.C., N.C.B. and S.J.M-M. designed research; K.B., G.P.M., W.S.E., G.P.C., J.G.F., E.R.A., A.D., L.C.G., S.J.P., R.D., M.K., L.D.G., L.E., F.S., S.E.M., G.P. and C.S.-K. performed research; K.B., I.P., E.R.A., A.D., Y.L., P.D., R.D., S.M.C., N.C.B. and S.J.M.-M. analyzed data; K.B. and S.J.M.-M. wrote the manuscript with contributions from co-authors. All authors reviewed and approved the final manuscript.

### Additional information

**Competing interests:** The authors declare no competing interests.

