## [Peer Review File · Nature Communications]

Reviewers' comments:

Reviewer #1 (Remarks to the Author):

Review Overview:

This is a significant research contribution, and represents a break-through in the understanding of how wood is deconstructed by marine animals. It also contributes to our general understanding of the mechanisms involved lignocellulose degradation in general across the tree of life. Despite the importance of the research in highlighting the new mechanism for lignocellulose deconstruction described, a concern is that the authors take the discussion beyond the level that the data support and extend their observations to describing hemocyanin as a new type of enzyme. As described below in the specific comments, more data is needed to demonstrate that an enzymatic-type reaction is occurring, and this data is not provided in the manuscript.

Prior research from 25 years ago demonstrated that hemoglobin could effectively delignify wood pulp in a catalytic manner, and that larger molecular weight lignin condensation products were formed. The hemoglobin was not discussed as an "enzyme" at that time, and it was described as catalytic action. It is quite interesting therefore that the copper-containing analogue of hemoglobin is now proposed as active agent in the deconstruction of lignin as part of a life process. The procedures and results, (which are appropriate and represent an important contribution to the literature) are not in need of significant revision. The discussion carefully describes the important findings and outlines the significance of the discovery relative to the context of lignocellulosic degradation. It is very important however, that the authors not overstep the data relative to the discussion of the type biochemical reaction going on. If future data support a discussion of hemocyanin having "enzymatic" activity, then discussion of naming protocols with IUBMB and IUPAC might be appropriate prior to publication.

This manuscript, upon modification, will provide important insight into marine animal biodegradation of lignocellulose and describes an important role for hemocyanins in this process.

Specific comments:

95-96: It is important not to perpetuate older information about brown rot fungal degradation of wood that is no longer accurate. Brown rot fungi completely depolymerize lignin, modify the monomers produced, and then re-polymerize it as extensively modified brown rotted lignin. Although it appeared to researchers for many years that the lignin was only slightly modified, this is not the case as it is in a new polymerized form. The key difference between white rot and brown rot is that white rot depolymerize, and then metabolize lignin. The brown rots depolymerize lignin, but then repolymerize the modified lignin and do not metabolize it. The sentences in this section should be revised to reflect this.

108: It is important to distinguish between deconstruction of lignin and metabolism of lignin in the wording of this sentence. Lignin undergoes "substantial breakdown" in the termite gut, through yet unknown mechanisms. The wording needs to be revised to indicate that lignin is deconstructed and modified, but not metabolized by termites.

128: Reword: Quinones can form polymers, but typically this would occur during redox cycling when a semi-quinone radical is produced. To say the quinone itself is forming polymers is a little misleading.

150-153: From the data presented, the majority of the hemicellulose components appear to have been digested in the "cellulose" fraction (Figure 1d) which does not agree with the generalization in this sentence. It may be that the hemicellulose that is more closely associated with paracrystalline cellulose is more readily depolymerized and digested; but the hemicellulose that is more closely associated with lignin is not. The sentence(s) should at least be modified to clarify that the information in Figure 1d is different than stated.

182-186: The authors clearly demonstrate cellulose removal, but dance around the modification of lignin (and hemicellulose) which must occur if cellulase enzymes are to access the cellulose for depolymerization and removal. As occurs in other organisms that digest woody substrates, the lignin must either be solubilized (and then metabolized), or it is repolymerized after depolymerization and is not metabolized. Modification of lignin is the only explanation of how a fecal pellet containing large amounts of lignin and reduced cellulose could be produced as the cellulose is essentially encased by the lignin. I suggest rather than vaguely stating “any modification of these polymers”, a clear statement be inserted that discusses how lignin must be modified to remove it from cellulose, and then deposited in solid form in a fecal pellet. This would not be minor modification of the lignin, but rather a significant modification.

261-300 (and Fig. 4): The wording of this section is concerning: An enzyme has a k_{cat} value, with a rate of conversion being involved. For this manuscript, conversion of a substrate (lignin) would need to occur over time to demonstrate enzymatic activity. Although this may be occurring, and may be demonstrated in subsequent work, the data presented does not demonstrate that. (Data is also not provided to demonstrate that hemocyanin is being consumed or altered by the reaction which would show that a non-enzymatic chemical reaction occurred as an alternative to an enzymatic reaction). Figure 4a shows only a single time point for hemocyanin conversion of pyrogallol, whereas for laccase, 3 time points are used for comparison. With the data presented, a simple chemical reaction may be occurring. To use the term “enzyme” to describe hemocyanin, or demonstrate that it is catalyzing an enzymatic-like reaction, would require that further kinetic studies be conducted to show at a minimum the enzymatic reaction rate. The current findings in and of themselves are quite valuable relative to describing a new mechanism for deconstruction of lignocellulose, but with the data provided it would be best to describe the reaction as chemical conversion. Discussion of an enzymatic reaction would not be valid unless, and until, a catalytic rate can be demonstrated. Inclusion of values such as K_m or V_{max} would be useful if an enzyme-like catalysis can be demonstrated.

303 -Figure 4a: The figure caption for 4a should be revised, as hemocyanin was not tested (or data is not shown) for time points other than 30 minutes, and laccase was not tested at 30 minutes.

371-373 and Supplemental Figure 7a: Discussion and supplemental data are provided on “repeated use” of hemocyanin which could potentially provide useful information relative to catalytic action. No information is provided however, on how the hemocyanin was recycled and more detail on the procedures for the data generation are needed if this information is to be reported in the manuscript. Was a portion of unreacted hemocyanin from an earlier experiment simply used in a new reaction, and if so, can this really be considered as a recycling of the enzyme? This should be clarified, or the information in 371-373 and supplemental figure 7a removed. A more traditional way of assessing enzymatic “repeated use” would be to provide k_{cat} values or some type of rate value.

382-383: Although intriguing, it would be overstepping the data at this point to discuss these as enzymes. Certainly though, it is still very important as a chemical reaction even if not a sustained catalytic reaction, as it does the needed “pretreatment”.

390-392: This sentence should be modified as there are many microorganisms that do not metabolize xylose sugars well.

423: The discussion of, “in a manner similar to laccases” should be deleted as it has not been demonstrated that the action is similar to laccase, nor that it is enzymatic. Further, the action of laccase on lignin requires a natural mediator to produce a radical which attacks lignin. The manuscript does not go into discussion of what compounds may function as mediators for hemocyanin, if this is even feasible or occurring in the limnoid system. It is better therefor to

leave mechanistic statements on "similar to laccases" out until work in this area has been done. Curiously, the manuscript discusses the need for laccase mediators in line 437, but does not relate that to any possible hemocyanin requirement for a mediator. This suggests that considerable work would be needed to classify hemocyanin as an enzyme, especially an enzyme similar to laccase.

Other lines that should be modified in the paper relative to the discussion of hemocyanins as enzymes include in the abstract, introduction and line 442.

Reviewer #2 (Remarks to the Author):

The manuscript reports for the first time that the hemolymph oxygen-carrier hemocyanin of a marine woodborer modifies lignin thereby improving digestability of cellulose contained in lignocellulosic plant biomass. The authors suggest that hemocyanin thus represents a new class of ligninases, which allow digestion of wood in the absence of gut symbionts.

To begin with I have to inform that I have no expertise on the digestion of cellulose/lignin, thus I am not in an position to comment on biochemical assays used to measure cellulose/lignin degradation.

However, I have worked for years with hemocyanin, type 3 copper proteins and related hemolymph proteins.

In the past decade it has emerged that hemocyanin besides being an oxygen-carrier protein in fact is a very multifunctional protein, which can produce antimicrobial peptides, elicit phenoloxidase activity, transport hormones, be a building of cuticule, just to name a few.

In this context I am not surprised about new activity of hemocyanin reported in this manuscript.

In my opinion the results are a new and important advance in the field of respiratory proteins and type 3 copper proteins and probably also give new impulses for biotechnological applications for lignin/cellulose degradation. As such the results merit publication in Nature Communications.

The paper is very well written in a clear and comprehensible style.

While I can't specifically comment on the experiments made to study cellulose/lignin degradation, when reading these experiments I did not notice obvious contradictions or illogical conclusions.

With respect to hemocyanin the authors purified hemocyanin for their experiments by whole body homogenization of animals. Normally this is not the method of choice. Given the manifold of enzymes present in the whole body, the hemocyanin sample could easily be contaminated by other enzymes which could elicit the enzymatic activity which is examined.

However, due to the small size of the animals whole body homogenization was most probably the best the authors could do. I have worked myself with crustaceans of the same small size and experienced that it can be impossible to withdraw sufficient hemolymph samples for purification. Recombinant expression of hemocyanin in order to produce a very pure hemocyanin sample is impossible, since hemocyanins have not been successfully recombinantly expressed in native form until now. Thus at this moment using whole body homogenates further purified by chromatography was most probably the best option.

In conclusion, if there are no problems concerning the cellulose/lignin part, the manuscript in my opinion can be published as is.

Reviewer #3 (Remarks to the Author):

Besser and co-authors describe their discovery of a new group of lignin-active enzymes, namely hemocyanins. The manuscript provides a comprehensive overview of lignocellulose digestion by marine woodborers of the genus *Limnoria*, and report the selective degradation of cellulose in wood fiber from willow. The authors do not observe lignin degradation by *Limnoria*, but do observe comparatively high expression of hemocyanin in fluids from the hideout of the wood borer, and lignin modification in fecal samples (e.g., more oxidized or condensed lignin as detected by ATR-FTIR).

The hemocyanin was partially purified from the hindgut and then used to treat alkali lignin. In short, corresponding ¹³C solid state NMR analysis of the resulting products suggested the hemocyanin may lead to demethoxylation of the lignin (line 286). The impact of copper chelators on this activity was investigated, as was the impact of hemocyanin on cellulase activity.

While the manuscript is clearly written, the claims regarding hemocyanin action on lignin requires further confirmation. In particular, the authors are asked to address the following specific points:

1. The authors use only one lignin source - alkali lignin. One would expect alkali lignin to be sulfonated. Please indicate the purity and provide a full compositional analysis of the lignin source (e.g., ash content, sulfur, carbohydrate content, G/S ratio). The authors should also include other lignin sources (e.g., ideally milled wood lignins)
2. Based on the carbon NMR analyses, the authors propose hemocyanins modify lignin through demethoxylation (line 286). This proposal should be tested by evaluating (a) the impact of hemocyanins on G-lignin and H-lignin from coniferous wood and agricultural fiber, respectively. Moreover, this should be directly investigated (e.g., by mass spectrometry) using corresponding monolignol compounds. Also, how do the authors reconcile this interpretation with the interpretation of ATR-FTIR, where they conclude the lignin may be more oxidized (line 179)?
3. The authors use SDS to activate the hemocyanin activity. Please clarify what could serve this function in nature (i.e., likely source of natural surfactant).
4. The impact of hemocyanins on cellulase action requires further development. For example, what is the affinity of hemocyanin to different lignins? Do these proteins bind lignin (in their functional form), which could reduce non-productive binding of cellulases, thereby promoting cellulase action? Also, how could lignin demethoxylation boost cellulase action? The authors also indicate that FTIR analyses suggest more condensed lignin structures after hemocyanin treatment. I would think this would reduce cellulase access to embedded cellulose fiber. If the authors think hemocyanins increase cellulase action through increasing their accessibility to cellulose, then an independent measure of cellulose accessibility should be obtained. For example, a measure of increased fiber porosity through nitrogen adsorption measurements or application of Simon staining.
5. The authors note the potential contribution of contaminating compounds with the native hemocyanin preparation, which could contribute to the reported activity. Given the potential impact of claims made in this study, studies of recombinant hemocyanin appear warranted.
6. Lastly, are hemicelluloses fully retained when feeding *Limnoria* sp. with coniferous wood fibre, which would indicate expression of highly specific endoglucanases

Reviewer #1 (Remarks to the Author):

Review Overview:

This is a significant research contribution, and represents a break-through in the understanding of how wood is deconstructed by marine animals. It also contributes to our general understanding of the mechanisms involved lignocellulose degradation in general across the tree of life. Despite the importance of the research in highlighting the new mechanism for lignocellulose deconstruction described, a concern is that the authors take the discussion beyond the level that the data support and extend their observations to describing hemocyanin as a new type of enzyme. As described below in the specific comments, more data is needed to demonstrate that an enzymatic-type reaction is occurring, and this data is not provided in the manuscript.

Prior research from 25 years ago demonstrated that hemoglobin could effectively delignify wood pulp in a catalytic manner, and that larger molecular weight lignin condensation products were formed. The hemoglobin was not discussed as an “enzyme” at that time, and it was described as catalytic action. It is quite interesting therefore that the copper-containing analogue of hemoglobin is now proposed as active agent in the deconstruction of lignin as part of a life process. The procedures and results, (which are appropriate and represent an important contribution to the literature) are not in need of significant revision. The discussion carefully describes the important findings and outlines the significance of the discovery relative to the context of lignocellulosic degradation. It is very important however, that the authors not overstep the data relative to the discussion of the type biochemical reaction going on. If future data support a discussion of hemocyanin having “enzymatic” activity, then discussion of naming protocols with IUBMB and IUPAC might be appropriate prior to publication.

This manuscript, upon modification, will provide important insight into marine animal biodegradation of lignocellulose and describes an important role for hemocyanins in this process.

Specific comments:

95-96: It is important not to perpetuate older information about brown rot fungal degradation of wood that is no longer accurate. Brown rot fungi completely depolymerize lignin, modify the monomers produced, and then re-polymerize it as extensively modified brown rotted lignin. Although it appeared to researchers for many years that the lignin was only slightly modified, this is not the case as it is in a new polymerized form. The key difference between white rot and brown rot is that white rot depolymerize, and then metabolize lignin. The brown rots depolymerize lignin, but then repolymerize the modified lignin and do not metabolize it. The sentences in this section should be revised to reflect this.

We thank the reviewer for this clarification. We have corrected the text of the manuscript accordingly (lines 96 to 99) and included an additional reference in support of this (Yelle *et al.* 2008).

108: It is important to distinguish between deconstruction of lignin and metabolism of lignin in the wording of this sentence. Lignin undergoes “substantial breakdown” in the termite gut, through yet unknown mechanisms. The wording needs to be revised to indicate that lignin is deconstructed and modified, but not metabolized by termites.

We have revised the text in lines 111 to 113 in line with this comment to distinguish modification and metabolism of lignin properly.

128: Reword: Quinones can form polymers, but typically this would occur during redox cycling when a semi-quinone radical is produced. To say the quinone itself is forming polymers is a little misleading.

We have addressed this point and rephrased the paragraph accordingly (lines 132 to 133).

150-153: From the data presented, the majority of the hemicellulose components appear to have been digested in the “cellulose” fraction (Figure 1d) which does not agree with the generalization in this sentence. It may be that the hemicellulose that is more closely associated with paracrystalline cellulose is more readily

depolymerized and digested; but the hemicellulose that is more closely associated with lignin is not. The sentence(s) should at least be modified to clarify that the information in Figure 1d is different than stated. We have modified the text (lines 155 to 161) describing the data in Figure 1 as suggested by the reviewer to reflect the results more accurately and to clarify the findings. Figure 1 has also been rearranged by swapping graph c and d to align with the data description in the modified text.

182-186: The authors clearly demonstrate cellulose removal, but dance around the modification of lignin (and hemicellulose) which must occur if cellulase enzymes are to access the cellulose for depolymerization and removal. As occurs in other organisms that digest woody substrates, the lignin must either be solubilized (and then metabolized), or it is repolymerized after depolymerization and is not metabolized. Modification of lignin is the only explanation of how a fecal pellet containing large amounts of lignin and reduced cellulose could be produced as the cellulose is essentially encased by the lignin. I suggest rather than vaguely stating “any modification of these polymers”, a clear statement be inserted that discusses how lignin must be modified to remove it from cellulose, and then deposited in solid form in a fecal pellet. This would not be minor modification of the lignin, but rather a significant modification.

We have modified the text (lines 197 to 206) to clarify this and included an additional reference (Simmons *et al.* 2016).

261-300 (and Fig. 4): The wording of this section is concerning: An enzyme has a k_{cat} value, with a rate of conversion being involved. For this manuscript, conversion of a substrate (lignin) would need to occur over time to demonstrate enzymatic activity. Although this may be occurring, and may be demonstrated in subsequent work, the data presented does not demonstrate that. (Data is also not provided to demonstrate that hemocyanin is being consumed or altered by the reaction which would show that a non-enzymatic chemical reaction occurred as an alternative to an enzymatic reaction). Figure 4a shows only a single time point for hemocyanin conversion of pyrogallol, whereas for laccase, 3 time points are used for comparison. With the data presented, a simple chemical reaction may be occurring. To use the term “enzyme” to describe hemocyanin, or demonstrate that it is catalyzing an enzymatic-like reaction, would require that further kinetic studies be conducted to show at a minimum the enzymatic reaction rate. The current findings in and of themselves are quite valuable relative to describing a new mechanism for deconstruction of lignocellulose, but with the data provided it would be best to describe the reaction as chemical conversion. Discussion of an enzymatic reaction would not be valid unless, and until, a catalytic rate can be demonstrated. Inclusion of values such as K_m or V_{max} would be useful if an enzyme-like catalysis can be demonstrated.

We have edited the text throughout to reflect the reviewer’s comments regarding the use of the term enzyme/enzymatic in conjunction with hemocyanin (lines 290/291, 296, 311, 317).

303 -Figure 4a: The figure caption for 4a should be revised, as hemocyanin was not tested (or data is not shown) for time points other than 30 minutes, and laccase was not tested at 30 minutes.

The figure 4 a and legend have been modified to include the time course for hemocyanin activity on pyrogallol (lines 325-327).

371-373 and Supplemental Figure 7a: Discussion and supplemental data are provided on “repeated use” of hemocyanin which could potentially provide useful information relative to catalytic action. No information is provided however, on how the hemocyanin was recycled and more detail on the procedures for the data generation are needed if this information is to be reported in the manuscript. Was a portion of unreacted hemocyanin from an earlier experiment simply used in a new reaction, and if so, can this really be considered as a recycling of the enzyme? This should be clarified, or the information in 371-373 and supplemental figure 7a removed. A more traditional way of assessing enzymatic “repeated use” would be to provide k_{cat} values or some type of rate value.

We have added the details of the re-use procedures in Material and Method section headed “Digestibility assays” (lines 898 to 902) and explained the process in the results section (lines 430 to 433).

382-383: Although intriguing, it would be overstepping the data at this point to discuss these as enzymes. Certainly though, it is still very important as a chemical reaction even if not a sustained catalytic reaction, as it does the needed “pretreatment”.

We changed the wording from “...lignin active enzymes...” to ““...lignin active proteins...” (line 444)

390-392: This sentence should be modified as there are many microorganisms that do not metabolize xylose sugars well.

We have corrected the sentence by adding “many symbiotic” (microbes.... line 452)

423: The discussion of, “in a manner similar to laccases” should be deleted as it has not been demonstrated that the action is similar to laccase, nor that it is enzymatic. Further, the action of laccase on lignin requires a natural mediator to produce a radical which attacks lignin. The manuscript does not go into discussion of what compounds may function as mediators for hemocyanin, if this is even feasible or occurring in the limnoid system. It is better therefor to leave mechanistic statements on “similar to laccases” out until work in this area has been done. Curiously, the manuscript discusses the need for laccase mediators in line 437, but does not relate that to any possible hemocyanin requirement for a mediator. This suggests that considerable work would be needed to classify hemocyanin as an enzyme, especially an enzyme similar to laccase.

We removed the mechanistic statement with reference to laccase (line 485) to address these concerns.

Other lines that should be modified in the paper relative to the discussion of hemocyanins as enzymes include in the abstract (we have changed line 54 accordingly), introduction (there is no mention of the term enzyme, we only refer to ligninolytic activity in line 139) and line 442 (we changed “enzymes” to “proteins” in line 504). We also edited line 60 in the significance statement (“lignin-modifying enzymes “ to “method for lignin modification”) and line 430 (ligninolytic “enzymes” to “agents”) to reflect the reviewer’s comments.

We thank Reviewer #1 for all comments and suggestions.

Reviewer #2 (Remarks to the Author):

The manuscript reports for the first time that the hemolymph oxygen-carrier hemocyanin of a marine woodborer modifies lignin thereby improving digestability of cellulose contained in lignocellulosic plant biomass. The authors suggest that hemocyanin thus represents a new class of ligninases, which allow digestion of wood in the absence of gut symbionts.

To begin with I have to inform that I have no expertise on the digestion of cellulose/lignin, thus I am not in an position to comment on biochemical assays used to measure cellulose/lignin degradation.

However, I have worked for years with hemocyanin, type 3 copper proteins and related hemolymph proteins.

In the past decade it has emerged that hemocyanin besides being an oxygen-carrier protein in fact is a very multifunctional protein, which can produce antimicrobial peptides, elicit phenoloxidase activity, transport hormones, be a building of cuticle, just to name a few.

In this context I am not surprised about new activity of hemocyanin reported in this manuscript.

In my opinion the results are a new and important advance in the field of respiratory proteins and type 3 copper proteins and probably also give new impulses for biotechnological applications for lignin/cellulose degradation. As such the results merit publication in Nature Communications.

The paper is very well written in a clear and comprehensible style.

While I can't specifically comment on the experiments made to study cellulose/lignin degradation, when reading these experiments I did not notice obvious contradictions or illogical conclusions.

With respect to hemocyanin the authors purified hemocyanin for their experiments by whole body homogenization of animals. Normally this is not the method of choice. Given the manifold of enzymes present in the whole body, the hemocyanin sample could easily be contaminated by other enzymes which could elicit the enzymatic activity which is examined.

However, due to the small size of the animals whole body homogenization was most probably the best the authors could do. I have worked myself with crustaceans of the same small size and experienced that it can be impossible to withdraw sufficient hemolymph samples for purification. Recombinant expression of hemocyanin in order to produce a very pure hemocyanin sample is impossible, since hemocyanins have not been successfully recombinantly expressed in native form until now. Thus at this moment using whole body homogenates further purified by chromatography was most probably the best option.

In conclusion, if there are no problems concerning the cellulose/lignin part, the manuscript in my opinion can be published as is.

We thank Reviewer #2 for these comments.

Reviewer #3 (Remarks to the Author):

Besser and co-authors describe their discovery of a new group of lignin-active enzymes, namely hemocyanins. The manuscript provides a comprehensive overview of lignocellulose digestion by marine woodborers of the genus *Limnoria*, and report the selective degradation of cellulose in wood fiber from willow. The authors do not observe lignin degradation by *Limnoria*, but do observe comparatively high expression of hemocyanin in fluids from the hideout of the wood borer, and lignin modification in fecal samples (e.g., more oxidized or condensed lignin as detected by ATR-FTIR).

The hemocyanin was partially purified from the hindgut and then used to treat alkali lignin. In short, corresponding ¹³C solid state NMR analysis of the resulting products suggested the hemocyanin may lead to demethoxylation of the lignin (line 286). The impact of copper chelators on this activity was investigated, as was the impact of hemocyanin on cellulase activity.

We would like to emphasize that our data do not support specific demethoxylation of lignin necessarily, but that we find relative reduction in signals both for aromatic rings and methoxyl carbons which suggests that methoxylated aromatic rings are targeted by hemocyanin activity (line 305-307: "...the simultaneous relative reduction of the O-aromatic and aryl methoxyl carbon signals, ..., suggests that hemocyanin acts mostly on the aromatic-OCH₃ sites." originally line 286). If the aromatic rings were cleaved, the aryl methoxy group could change into an aliphatic side group without being removed itself, which could be responsible for the signal appearing at 24.9 ppm in the spectrum in Fig 4 b.

While the manuscript is clearly written, the claims regarding hemocyanin action on lignin requires further confirmation. In particular, the authors are asked to address the following specific points:

1. The authors use only one lignin source - alkali lignin. One would expect alkali lignin to be sulfonated. Please indicate the purity and provide a full compositional analysis of the lignin source (e.g., ash content, sulfur, carbohydrate content, G/S ratio). The authors should also include other lignin sources (e.g., ideally milled wood lignins)

We apologize for omitting details on the lignin used; we used Sigma #471003 Lignin, alkali, which is of low sulfonate content, containing 3.3% sulfur and 50.1% carbon as stated by the manufacturer. This is now included in Material and Methods (lines 870- 871). We have not performed a full compositional analysis as it seems to be a standard lignin source used in many research projects.

To test many other lignin sources would be extremely difficult with respect to the challenge of producing enough hemocyanin from its native source. This would require many 100s of animals to be sacrificed to extract the amount of hemocyanin required. We had to prioritize the use of hemocyanin (produced from 700 animals) to address other questions raised by the reviewers. The fact that hemocyanin can modify alkali lignin and is efficient in the pretreatment of lignocellulose from hard- and softwood sources (see below), but has no effect in pretreating cellulose alone, strongly argues for a ligninolytic activity of hemocyanin to enable access of cellulases to their substrate. We felt it was more productive to use the available hemocyanin for other suggested experiments to investigate binding of hemocyanin to the biomass and the impact on fibre porosity (see below under 4.).

2. Based on the carbon NMR analyses, the authors propose hemocyanins modify lignin through demethoxylation (line 286). Please see our first comment above. This proposal should be tested by evaluating (a) the impact of hemocyanins on G-lignin and H-lignin from coniferous wood and agricultural fiber, respectively.

Here we show data for improved digestibility of Scots pine sapwood after pretreatment with hemocyanin or seawater using the same conditions as for the willow experiments: 10 min pretreatment followed by 6 h saccharification with *Limnoria* GH7. The digestibility is similarly increased in softwood compared to hardwood. We have added these data to Supplementary Fig. 8 (as new part c, see figure below) mentioned in lines 359 to 360 in the main manuscript and amended the Materials & Methods section accordingly (line 890).

new Supplementary Fig. 8 (c)

Moreover, this should be directly investigated (e.g., by mass spectrometry) using corresponding monolignol compounds.

Our data do not necessarily support evidence for specific demethoxylation of lignin (s.a.). The ssNMR on lignin show the combined changes of methoxyl and aromatic signals (Fig. 4), if there was increased demethoxylation independent of aromatic ring cleavage, one could expect to see this in the difference spectrum. To make this clearer, we included the difference spectra for the ssNMR data of lignin treated with hemocyanin, which show simultaneous reduction in both regions (see modified Fig 4. b insets and lines 306-307 in manuscript). In addition, the difference ssNMR spectrum of wood and fecal pellets scaled to the methoxy lignin peak at 56ppm leaves a spectrum consistent with cellulose (and xylan), but no other peaks (or troughs) in the various lignin regions become apparent (Suppl Fig. 2).

The exact mechanism of the hemocyanin activity is not elucidated and fully understood, but it can likely be expected to be different towards polymeric and monomeric substrates. We tested about 30 phenolic compounds for hemocyanin activity in the UV-vis assay, including p-coumaric acid, trans-ferulic acid and coniferyl alcohol, but failed to detect changes in the spectra of these monolignolic compounds.

Also, how do the authors reconcile this interpretation with the interpretation of ATR-FTIR, where they conclude the lignin may be more oxidized (line 179)?

The manuscript states: "Attenuated total reflectance (ATR)-FTIR spectra are consistent with, an increase in lignin (1630-1670, 1160 and 1140 cm^{-1}), which may be more oxidized or condensed (1640 and 1550 cm^{-1}), in fecal pellets in comparison to wood". According to prior publications, the stated region (1640 and 1550 cm^{-1}) is associated with lignin and oxidized lignin structures. On revision we would like to remove "or condensed" (line 194) as the cited wavenumbers do not refer to condensed structures.

The observed activity of hemocyanin towards soluble lignin may indicate a general property of hemocyanins to attack the lignin component of lignocellulose, but the *in vitro* experiment is only an approximation of the reaction *in vivo*, as soluble lignin is not the same as native lignin (which is also present in a complex and not pure), and not all components present in the animal's digestive system that could contribute to the observed modification of wood during digestion are known and therefore potentially not included in the *in vitro* reaction. The FTIR data are a comparative analysis of wood and fecal pellets, which had been derived from the "*in vivo*" digestion process.

3. The authors use SDS to activate the hemocyanin activity. Please clarify what could serve this function in nature (i.e., likely source of natural surfactant).

SDS is an artificial activator routinely used in assays for detection of PO activity of hemocyanins (Jaenicke *et al.* 2009). It is assumed that the presence of SDS in micellar concentrations mimics the effects of natural activators in arthropods (such as lipoproteins, phospholipids, proteases and small antimicrobial peptides) and interacts with hemocyanin without inducing denaturation, but inducing conformational change that exposes the entrance to the active site for potential substrates. Cong *et al.* (2009) provided structural evidence for this mechanism.

In the digestive system of *Limnoria*, the conformational changes of hemocyanin appears to be facilitated by seawater, as we see a notable thermal shift of the melting profile of the protein in seawater compared to buffer (Suppl. Fig. 7 a - c, copied below); the natural activator does not need to be a surfactant as such.

4. The impact of hemocyanins on cellulase action requires further development. For example, what is the affinity of hemocyanin to different lignins? Do these proteins bind lignin (in their functional form), which could reduce non-productive binding of cellulases, thereby promoting cellulase action?

We carried out new experiments to address these questions; willow wood powder was incubated with hemocyanin or seawater as previously described or with BSA (under same conditions/concentration as hemocyanin), then these pretreatment solutions were removed and the pretreated wood was incubated with the cellobiohydrolase also as described in the manuscript (*Hypocrea* CBH I or *Limnoria* GH7). The presence of proteins was then assessed by SDS-PAGE in the wood pellets after saccharification and in the hydrolysate in comparison to the pretreatment (PT) and saccharification (Sacch) solutions prior to incubation (aliquots of 16 μ L of solutions plus 4 μ L 5x loading buffer and wood pellets plus 100 μ L 1x loading buffer were incubated 5 min at 100 $^{\circ}$ C, then 20 μ L each loaded onto gel). Gels with samples from *Hypocrea* CBH I assay were stained with Coomassie Instant blue stain (see figure below, A), gels with samples from *Limnoria* GH7 assay were blotted for Western analysis, stained with Ponceau and probed with anti-GH7 antibodies (see figure below, B).

A – Coomassie stained SDS-PAGE gel (new Supplementary Fig. 9):

Result A: a small proportion of hemocyanin remained in the wood pellet after pretreatment, the pretreatment with hemocyanin had no impact on the binding of the cellulase to the wood and the majority of the cellulase remained in solution.

B – Ponceau-stained blot (top) and Western blot (bottom)

Result B: a small proportion of hemocyanin and BSA remained in the wood pellet after pretreatment, some hemocyanin and BSA was released into the hydrolysate during saccharification. The *Limnoria* cellulase (no CBM) did not bind to the wood under any PT condition and a similar amount of GH was detected in the hydrolysate and the saccharification solution prior to application in hemocyanin and seawater treated wood, less GH remained in the hydrolysate after BSA pretreatment.

Both experiments (A and B) show that hemocyanin does not bind extensively to lignocellulose and that hemocyanin pretreatment does not seem to alter the binding of cellulases to lignocellulose.

We present the data from experiment A in a new Supplementary Figure 9, with a description in the manuscript (lines 380 to 389) and in Materials & Methods (lines 913 to 924).

Also, how could lignin demethoxylation boost cellulase action? The authors also indicate that FTIR analyses suggest more condensed lignin structures after hemocyanin treatment. I would think this would reduce cellulase access to embedded cellulose fiber. If the authors think hemocyanins increase cellulase action through increasing their accessibility to cellulose, then an independent measure of cellulose accessibility should be obtained. For example, a measure of increased fiber porosity through nitrogen adsorption measurements or application of Simon staining.

As mentioned above we have removed the term “condensed” from our interpretation of the FT-IR data of wood and fecal pellets as it was used incorrectly (line 194). It is often stated in the literature that depolymerisation followed by re-polymerisation can lead to more condensed lignin, which may describe a modification of lignin itself and of its distribution in the biomass.

However, to get an idea about the porosity of lignocellulose and an assessment of accessibility to cellulose, we have now carried out NMR T_2 relaxation time measurements of hemocyanin pretreated wood powder. We incubated ten 10 mg willow samples with either seawater or 100 μg hemocyanin each for 3 h at room temperature and removed solutions after pelleting of biomass which was then dried, pooled and analysed according to the following methodology.

Samples were saturated with N,N-Dimethylacetamide (DMAc) and any excess removed. DMAc penetrates the pores in the material and its relaxation time T_2 becomes relative to the size of the pore it is filling (the smaller the pore, the shorter T_2) and to the pore surface ability to produce spin relaxation of the DMAc protons (so-called surface relaxivity parameter). This parameter depends on surface properties such as magnetic susceptibility and affinity to the enclosed fluid. The T_2 relaxation time is measured as the decay rate of echoes obtained in the so called Carr-Purcell-Meiboom-Gill (CPMG) experiments. In realistic scenarios, the distribution of pore sizes and differences in the fluid affinity within the pores result in a multiexponential decay of the CPMG signal, i.e. a distribution of T_2 times, which can be recovered using a Laplace Inversion procedure (Borgia *et al.* 1998; Provencher 1982). Therefore, T_2 values can be converted to a length scale and reflect information about the presence of pores in scales from few nanometers to few micrometers. Because lignocellulose has characteristic interstitial scales (pores), the T_2 distribution is not homogeneous, but presents a profile with components that encode the length scales of the pores as well as their occupancy by the enclosed fluid (Tsuchida *et al.* 2014), i.e. DMAc in our case. The proportion of DMAc in each interstitial scale is estimated from the relative areas of the respective components on the T_2 distribution by fitting to a log-Gaussian function; this can be correlated to the accessibility of DMAc molecules in each type of pore.

When comparing willow wood pretreated with seawater (SW) or hemocyanin (Hc), it is apparent that the T_2 profiles of both samples are similar in shape, i.e. reveal the same type of pore distribution with three populations accessed by DMAc at similar length scales (**figure a, b below**). Assuming a similar surface relaxivity of DMAc in willow as in cotton cellulose, the length scales corresponding to these profiles are estimated to be in the order of 1 to 10 nm, 10 to 100 nm and 100 to 1000 nm (Zhang *et al.* 2016; **figure b below**). These pore populations are interpreted as interstitial spaces in the amorphous fraction of cellulose (surface of cellulose fibrils), voids in the inter-microfibril spaces, and small luminal structures, respectively (Tsuchida *et al.* 2014). However, the relative area of the two lower T_2 population curves increased considerably in the hemocyanin pretreated sample (**figure c below**), which indicates that there are more pores within these two length scales (both associated with the accessibility of the cell wall) after hemocyanin incubation. This suggests that there is increased porosity of lignocellulose and potentially better accessibility of cellulose by hydrolytic enzymes upon hemocyanin pretreatment.

We have presented the data from this experiment in a new Supplementary Figure 10, with a description in the manuscript (lines 391 to 413) and in Materials & Methods (lines 926 to 955), and have included lines 443/444 in the Discussion and all relevant references (#64-67, #103-108).

new Supplementary Figure 10

5. The authors note the potential contribution of contaminating compounds with the native hemocyanin preparation, which could contribute to the reported activity. Given the potential impact of claims made in this study, studies of recombinant hemocyanin appear warranted.

We agree that the recombinant expression of hemocyanins would be very useful, but this has been so far unsuccessful. We have attempted expression in different host systems over several years including *E. coli*, *Pichia pastoris*, insect cells, human embryonic kidney cells and *Aspergillus*, without success. We would also like to cite Reviewer 2: "Recombinant expression of hemocyanin in order to produce a very pure hemocyanin sample is impossible, since hemocyanins have not been successfully recombinantly expressed in native form until now. Thus at this moment using whole body homogenates further purified by chromatography was most probably the best option." We took great care to control as far as possible for any effect potentially derived from contaminants.

6. Lastly, are hemicelluloses fully retained when feeding *Limnoria* sp. with coniferous wood fibre, which would indicate expression of highly specific endoglucanases

We have data for compositional analysis of Scots pine sapwood and fecal pellets, for which a mass loss of 22 % (as for willow digestion) was determined. In Scots pine, most of the loss is also accounted for by the hydrolysis of the cellulosic fraction (40 % reduction; **see figure below, a**), which presents a release of mostly glucose and some mannose (**see figure below, b**). In contrast to willow, the hemicellulosic fraction is also reduced, although to a lesser extent than the cellulosic fraction by about 20 % (**see figure below, a**). Softwood hemicellulose consists mainly of galactoglucomannan, but also of some arabinoglucuronoxylan, and during digestion mostly mannose and some glucose are released from this fraction, with little loss of xylose content (**see figure below, c**). So overall, in softwood mainly glucose but also mannose seems to be removed by the woodborer, suggesting that polymers containing hexose sugars are the main targets for digestion.

We added this data to the manuscript as a new Supplementary Fig. 1 and a brief description in the main text (lines 162 to 169). Text edits in the Discussion related to this point include line 438 and 447.

new Supplementary Fig. 1

We thank Reviewer #3 for all comments and suggestions.

REVIEWERS' COMMENTS:

Reviewer #1 (Remarks to the Author):

The paper addresses the majority of my concerns, and is acceptable in my view. However because the mechanism appears to related to that of previously reported work with hemoglobin on lignin, it would be appropriate to cite older work that was done with hemoglobin on lignin and make brief reference to the related nature of the structures of hemoglobin and hemocyanin. See references going back to the 1980s by Pettersson, Ander, Eriksson, such as this one:

<https://link.springer.com/article/10.1007/BF00170196>

This would be done for example in line 444, where the language "new class of lignin active proteins" would be better modified to reference that prior synthetic work looked at the action of hemoglobin, but that this (current manuscript) is the first known discovery of a respiratory protein functioning in a biological system in this manner against lignin.

Reviewer #3 (Remarks to the Author):

The authors have carefully addressed each of my questions and concerns. I have no further comments.

REVIEWERS' COMMENTS:

Reviewer #1 (Remarks to the Author):

The paper addresses the majority of my concerns, and is acceptable in my view. However because the mechanism appears to related to that of previously reported work with hemoglobin on lignin, it would be appropriate to cite older work that was done with hemoglobin on lignin and make brief reference to the related nature of the structures of hemoglobin and hemocyanin. See references going back to the 1980s by Pettersson, Ander, Eriksson, such as this one:

<https://link.springer.com/article/10.1007/BF00170196>

This would be done for example in line 444, where the language “new class of lignin active proteins” would be better modified to reference that prior synthetic work looked at the action of hemoglobin, but that this (current manuscript) is the first known discovery of a respiratory protein functioning in a biological system in this manner against lignin.

While we can see the thematic connection, i.e. hemoglogin and hemocyanin are both best known for their oxygen transport roles, we think their differences may not be fully appreciated by the reviewer. The structure of the active sites of hemoglobins and hemocyanins are very different, hemoglobin has an iron-porphyrin (heme) oxygen binding site, whereas hemocyanin has a type-3 copper center. We also note that the hemoglogin activity is only seen in the presence of hydrogen peroxide and other oxidants, whereas no further oxidant is required in the case of hemocyanin.

We have included the suggested work and replaced the sentence in line 444 (now 355) by the following: “Ander *et al.* (1990 and references therein) reported the activity of hemoglobin and other heme compounds towards lignin in the presence of oxidants such as hydrogen peroxide, in a similar fashion to peroxidase-type ligninases. In contrast, we report the discovery that respiratory hemocyanins in the *Limnoria* hindgut enhance lignocellulose digestibility by a mechanism that requires no additional oxidant and appears to function in a manner distinct to that reported for hemoglobin.”

Reviewer #3 (Remarks to the Author):

The authors have carefully addressed each of my questions and concerns. I have no further comments.

Thank you.